



# Reviews and Syntheses: Carbon biogeochemistry of Indian estuaries

**Manab Kumar Dutta[1]\*, Krishnan Sreelash[1], Damodaran Padmalal[1], Nicholas D. Ward[2-3], Thomas S. Bianchi[4]**

[1]National Centre for Earth Science Studies (NCESS), Ministry of Earth Science, Government of India, PB no – 7250, Akkulam, Thiruvananthapuram – 695011, India

[2]Pacific Northwest National Laboratory, Marine and Coastal Research Laboratory, Sequim, WA98382 USA

[3]University of Washington, School of Oceanography, Seattle, WA 98195 USA

[4]University of Florida, Department of Geological Sciences, Gainesville, FL USA

\*_Corresponding author_: manabdutta.1987@gmail.com





## Abstract

The goal of this review is to provide a comprehensive overview of the magnitude and drivers of carbon cycling dynamics in the major estuaries of India. Data from a total of 32 estuaries along the Bay of Bengal (BB) and the Arabian Sea (AS) were compiled from the literature and re-analysed based on changes in season (wet vs. dry) and marine end-members (e.g., BB vs. AS). The estuaries are generally undersaturated in dissolved oxygen relative to the atmosphere and strongly influenced by local and regional precipitation patterns. Speciation of the dissolved inorganic carbon (DIC) pool is dominated by bicarbonate and primarily variability in DIC is controlled by a combination of carbonate weathering, the degree of precipitation, the length of the estuaries, in situ respiration, and mixing. Carbonate dissolution had the largest influence on DIC during the wet season, while respiration was the primary control of DIC variability in the estuaries connected with BB during the dry season. Interestingly, the influence of anaerobic metabolism on DIC is observed in the oxygenated mangrove dominated estuaries, which we hypothesize is driven by porewater exchange in intertidal sediments. Dissolved organic carbon (DOC) generally behaves non-conservatively in the studied estuaries. The DOC-particulate organic carbon (POC) inter-conversion and DOC mineralization are evident in the BB during the dry season and AS estuaries, respectively. The wet season $\delta^{13}C_{POC}$ shows dominance of freshwater algae, $C_3$ plant material, as well as marine organic matter in POC. However, anthropogenic inputs are evident in some estuaries in eastern India during the dry season. POC respiration was identified in the AS; however, a link between POC and $CH_4$ is identified throughout both the regions. $pCO_2$ is controlled principally by respiration with freshwater discharge only playing a marginal important role in the BB. The AS estuaries act as a $CO_2$ source to the atmosphere; however, the BB estuaries vary between a source and sink. POC together with methanotrophy and dam abundance appear to control $CH_4$ concentrations, and all of the studied estuaries act as a $CH_4$ source to the atmosphere. Additionally, anthropogenic inputs and groundwater exchange also show potential influences in some cases. The Indian estuaries contribute 2.62% and 1.09% to the global riverine DIC and DOC exports to the ocean, respectively. The total



$CO_2$ and $CH_4$ fluxes from Indian estuaries are estimated as ~9718 Gg $yr^{-1}$ and 3.27 Gg $yr^{-1}$,
which contributes ~0.67% and ~0.12%, respectively, to global estimates of estuarine
greenhouse gas emissions. While a qualitative idea on the major factors controlling the
carbon biogeochemistry in India is presented through this work, a more thorough
investigation including rate quantification of the above-mentioned mechanisms is essential
for precise accounting of the C budget of Indian estuaries.
**Keywords**
Carbon cycling, trace gases, estuary, mangroves, India










## Introduction


Estuaries, where inland waters mix with the coastal ocean, serve as important centres
of C cycling at the land-ocean interface (e.g., Bianchi et al 2018). These dynamic ecosystems
with abundant biodiversity and biological activity are emerging as a net source of carbon
dioxide ($CO_2$) and methane ($CH_4$) to the atmosphere as most of the world's large rivers and
estuaries are being reported to be oversaturated with respect to $CO_2$ and $CH_4$ (Bouillon et al.,
2003). A fraction of $CO_2$ removed from the atmosphere by terrestrial systems during
photosynthesis and weathering reactions is exported into rivers and estuaries as inorganic and
organic carbon; a significant portion of this exported C is ultimately recycled back into the
atmosphere (Ward et al., 2017). Although estuaries only cover ~4% of the continental shelf
regions, globally, the amount of $CO_2$ outgassed from estuaries is similar to $CO_2$ uptake in
continental shelf regions of the world, albeit with large uncertainty (Borges et al., 2003; Cai
et al., 2003; Cai and Wang, 1998). This suggests that estuaries are not only active pathways
for C transport (Bianchi and Bauer, 2011; Bauer and Bianchi, 2011; Dutta et al., 2019a) but
potentially a niche for labile OM modification by biogeochemical processes (Frankignoulle et
al., 1998; Bianchi 2011). In addition, surface run off, anthropogenic activities (including both
municipal as well as industrial) and groundwater inputs also contribute to the estuarine C
pool. Thereafter, based upon oxygen levels and residence time in the estuary, among other
factors, C undergoes complex biogeochemical transformations before transiting to the
continental shelf region and/or atmosphere.
Rivers are the major source of organic matter (OM) to the coastal environment as they
transport OM derived from vascular plants and soils from the terrestrial environment to the
ocean (Onstad et al., 2000; Li et al., 2017; Raymond et al., 2001). Lateral export from coastal
wetlands and subterranean groundwater discharge also deliver OM to estuaries, but these
fluxes remain poorly constrained (Moore and Joy, 2021; Santos et al., 2021). The terrestrial
OM derived from continental land masses is one of the major energy sources for aquatic and
marine organisms (Sedell et al., 1989; Wang et al., 2014; Krishna et al., 2015). Therefore,
riverine transport of OM is not only directly link with the global C cycle but also plays a





pivotal role in the food web dynamics of freshwater and coastal ecosystems (Caffrey, 2004).
Contributing ~66% of global river water discharge, tropical rivers deliver ~0.53 Pg C to
estuaries annually (Huang et al., 2012) of which dissolved organic C (DOC) and particulate
organic C (POC) contribute ~210 and 170 Tg C $yr^{-1}$, respectively (Ludwig et al., 1996).
Along with rivers, tidal vegetated wetlands (mangroves, salt marshes, and seagrass), also play
a significant role in the coastal ocean (<200 m water depth, covering ~7% of the ocean
surface) C budget (Bauer et al., 2013; Rosentreter et al., 2018). Very similar to the estuaries,
the tidal vegetated wetlands act as a lateral C filter as well as a hotspot for biogeochemical C
transformation (Koné and Borges, 2008, Nellemann et al., 2009; Mcleod et al., 2011;
Breithaupt et al., 2012, Regnier et al., 2013; Rosentreter et al., 2018; Dutta et al., 2019a).
Despite having immense biogeochemical significance, the tidal vegetated wetlands are
disappearing at an alarming rate on annual scale (mangroves: ~0.7 – 3%; seagrass: ~7%,
saltmarsh: ~1 – 2%; Mcleod et al., 2011). Therefore, comprehensive investigation of them is
needed to understand implication of ecosystem loss on coastal C biogeochemistry as well as
more precise accounting of global C budgets (Alongi, 2002; Mcleod et al., 2011; Bauer et al.,

2013).

While substantial insight has been gained on estuarine OC cycling and $CO_2$ exchange,
the magnitude of $CH_4$ emissions or uptake by estuaries remains poorly constrained. Aquatic
ecosystems account for nearly half of global emissions of $CH_4$ from natural and
anthropogenic sources; estuaries and coastal vegetated ecosystems only account for a small
amount of these aquatic emissions (Al-Haj and Fulweiler, 2020; Saunois et al., 2020).
However, biogenic $CH_4$ emissions from aquatic systems, including estuaries, are likely to
increase with increasing coastal urbanization and eutrophication (Rosentreter et al., 2021).
Further, the abundance of thermogenic sources of $CH_4$ in tectonically active estuarine
seafloors remain poorly documented and potentially large positive feedback for climate
change (Johnson et al., 2022).The coast of India is home to numerous and diverse estuarine
systems facing varying degrees of anthropogenic pressure; to date, studies of Indian estuaries
have largely focused on either single estuaries with wide spatial coverage (Mukhopadhyay et



al., 2006; Samanta et al., 2015; Dutta et al., 2019a, 2021; Gupta et al., 2008; Pattanaik et al.,
2017; Bhavya et al., 2017; Bouillon et al., 2003; Sarma et al., 2011), or a large number of
estuaries with limited sampling locations (Sarma et al., 2012, 2014; Krishna et al., 2015,
2019; Rao et al., 2016). Moreover, many of these estuaries have extensive coastal wetlands,
particularly mangroves, which are densely distributed in estuaries of the Sundarbans
(Saptamukhi, Thakuran, and Matla) and more sparsely scattered along the banks of the
Haldia, Mahanadi, Godavari, Krishna, Ponnayaar, Mandovi and Zuari Rivers (Dutta et al.,
2015; Rao et al., 2016).

In general, the C biogeochemistry of Indian estuaries bordered by extensive mangrove

systems, has received more attention than estuaries less influenced by mangroves (Biswas et
al., 2004, 2007; Mukhopadhyay et al., 2002; Dutta et al., 2013, 2015, 2017, 2019a, 2019b,
2021; Ray et al., 2011, 2013, 2015, 2018; Ganguly et al., 2011; Krithika et al., 2008; Borges
et al., 2003; Akhand et al., 2016, 2021). Some studies have focused on key drivers of OM
cycling (Sarma et al., 2012, 2014; Dutta et al., 2019a, 2021; Rao et al., 2016; Ray et al., 2011,
2013; Biswas et al., 2004, 2007; Mukhopadhyay et al., 2002), and others more on quantitative
assessments of C budgets (Dutta et al., 2013, 2015, 2017) and export fluxes (Ray et al., 2018;
Krishna et al., 2015, 2019). Nevertheless, there remain gaps in our knowledge on how
changes in seasons (wet vs. dry) and marginal seas (Bay of Bengal BB vs. Arabian Sea AS)
fit into the larger view of estuarine C processing in the dynamic coastal region. Additionally,
to our knowledge, there is no comprehensive review on C and trace gas ($CO_2$ & $CH_4$) cycling
of Indian major estuaries. In this review, the key objectives of this work are to: examine
differences in the major drivers of dissolved and particulate C biogeochemistry of the BB and
AS estuaries; understand basic differences in major controlling factors for the $CO_2$ and $CH_4$
cycling of the BB and AS estuaries; and establish the importance of major Indian estuaries in
regional and global C budgets.
**Study area**

The Indian sub-continent is located at the centre of the monsoon domain and

comprises three distinct zones—Peninsula, Indo-Gangetic alluvium and Extra-Peninsula—



each with distinct climatic and geo-environmental settings. The monsoon system is
significantly influenced by the orographic systems, which creates spatial disparity in the
monsoon rainfall across India. Global change in the monsoon system and hydrological
regime (Mathew et al., 2021) are inherently linked with C biogeochemistry of the Indian
estuaries.

The Indian Ocean includes the Arabian Sea, Laccadive Sea, Somali Sea, Andaman

Sea, and the Bay of Bengal, which collectively cover ~19.8% of the water on the Earth's
surface. The Indian Ocean is unique in terms of its geographic position as it is surrounded by
Asia to the north, Africa to the west, and Australia to the east. The Indian sub-continent
diverges the northern Indian Ocean into the Bay of Bengal (NE Indian Ocean) and Arabian
Sea (NW Indian Ocean) with ~5 times higher freshwater discharge to the former ($1.63 \times 10^{12}$
$m^3 \, yr^{-1}$) (Subramanian, 1993; Gauns et al., 2005). The large freshwater influx to the BB leads
to the development of a strong vertical salinity stratification that prevents vertical mixing
between nutrient rich subsurface water with the surface. Additionally, higher suspended
sediment loads limit the euphotic depth (Subramanian, 1993; Prasanna Kumar et al., 2002;
Madhupratap et al., 2003) thereby limiting the phytoplankton growth. The coupled interaction
limits productivity in the BB (Varkey et al., 1996; Prasanna Kumar et al., 2002). In contrast,
the strong upwelling together with convective mixing present in AS makes it as one of the
most productive regions in the world (Madhupratap et al., 1996; Muraleedharan and Prasanna
Kumar, 1996; Bhattathiri et al., 1996; Barber et al., 2001).

Freshwater discharge from Indian rivers is principally governed by the monsoon-

induced precipitation during the southwest (SW) monsoon (June – September, > 80% of its
annual rainfall; Soman and Kumar, 1990) with occasional rainfall during the northeast (NE)
monsoon (December - March) that is mostly stored in dam reservoirs for domestic, industrial
and irrigation uses. Due to minimal discharge during the non-monsoon period, the discharge
during the SW monsoon is considered to be roughly equivalent to the total annual discharge
for Indian rivers. The magnitude of discharge from these rivers depends on spatial variations
in rainfall over the catchment during SW monsoon with comparatively higher precipitation
along the SW (3000 mm) followed by the NE (1000 - 2500 mm), SE (300 - 500 mm) and





NW (200 - 500 mm) coasts of India (Soman and Kumar, 1990). Variability in discharge
changes the dominant source of organic matter inputs in Indian rivers (allochthonous or
autochthonous) and the contribution of these sources varies between estuaries depending on
river basin size, tidal amplitude, discharge characteristics, and water residence time. For
example, on the west coast the SW rivers drain red loamy soils in contrast to the NW rivers
that drain black soils. However, on the east coast, the rivers have red loamy and alluvial soils
in their upper and lower catchments, respectively except the Godavari and Krishna that
supply black soils in their upper catchment, red loamy and alluvial soils in their middle and
lower catchments, respectively (https://www.gsi.gov.in/). Changes in the source and nature of
C impacts the subsequent fate of C in the estuary. The diversity of terrestrial, freshwater, and
estuarine conditions across the Indian sub-continent makes it a particularly interesting setting
to evaluate varying drivers of estuarine C cycling. This diversity merits a thorough review on
the C biogeochemistry of Indian estuaries to highlight a holistic perspective of how Indian
estuaries serve as an integral part of the global C balance. The general characteristics of the
major Indian estuaries are presented in Table 1 and estuaries that have been included in this
review are presented in Figure 1.
**Material and methods**
The dissolved and particulate C as well as trace gas ($CO_2$ & $CH_4$) data from the
Indian estuaries were compiled and grouped according to the marginal sea they mix with (i.e.,
BB and AS). Similarly, data from wet (June – September; high freshwater discharge) and dry
(pre-monsoon: February – May & post-monsoon: October – January; low freshwater
discharge) seasons were pooled; the wet season has considerably more data than the dry.
Thus, mean pre- and post-monsoon data were considered to be dry seasons to improve
statistical rigor. All data collected from the literature were statistically reanalysed and
redrawn based on differences in wet/dry season and marginal sea end member using Sigma
plot Statistical Software V12. Statistical analysis showing $p < 0.05$ is considered statistically
significant while $p > 0.05$ was considered not significant. To highlight general features of the
Indian estuaries, estuaries having much scattered values compared to the others were





excluded from our re-analysis (see figures). Additionally, we have reassessed, recalculated,
and extrapolated the existing data wherever possible to extend quantitative understanding on
C budgets of the Indian estuaries as well as its impact on global C budgets. All the data used
in the paper is presented graphically in Fig. 2-6 and the correlations between parameters is
presented in the supplementary file (see Fig. S1-S20).
**Results**
*Salinity, dissolved oxygen, and pH variability*
In the dry season for BB and AS estuaries, surface water salinity ranged from 3.86 to
23.91 (mean: 12.49 ± 6.85) and 0.23 to 22.84 (mean: 11.96 ± 6.81), respectively. During the
wet season, salinity decreased more significantly in the AS estuaries (salinity: 0.04 – 7.32;
mean: 1.49 ± 2.53; 88% decrease) than the BB estuaries (salinity: 0.09 – 28.78; mean: 10.83
± 9.79; 13% decrease) (Fig.2A & 2B; Sarma et al., 2012; Rao et al., 2016; Dutta et al., 2015,
2019a, 2021; Samanta et al., 2015; Akhand et al., 2016; Ganguly et al., 2011; Pattanaik et al.,
2017). High salinities in both seasons are also associated with high tides (Akhand et al., 2016;
Dutta et al., 2019b).
Surface water %DO for the BB estuaries varied between 63 and 105% (mean: 93 ±
12%), 72 and 119% (mean: 95 ± 11%) during the wet and dry seasons, respectively, which is
higher than in the AS (wet season: 74 – 95%, mean: 85 ± 8%; dry season: 63 – 98%, mean:
81 ± 10%) (Sarma et al., 2012; Rao et al., 2016; Dutta et al. unpublished data; Fig. 2C & 2D).
Additionally, in vegetated wetlands bordering the estuaries, Dutta et al. (2019b) showed
lower %DO in a pre-monsoon diurnal C study in the Saptamukhi estuary during low tide.
Coinciding with %DO, BB estuaries had higher surface water pH values (wet season:
6.66 – 8.61, mean: 7.77 ± 0.52; dry season: 7.96 – 8.33, mean: 8.15 ± 0.12) than in AS during
both seasons (wet season: 5.98 – 7.51, mean: 6.84 ± 0.49; dry season: 7.23 – 7.90, mean: 7.70
± 0.31) (Fig. 2E & 2F; Sarma et al., 2012; Rao et al., 2016; Dutta et al., 2015, 2019a, 2021;
Samanta et al., 2015; Akhand et al., 2016; Ganguly et al., 2011; Pattanaik et al., 2017;
Bouillon et al., 2003; Piplode and Barde, 2015; Sangani and Manoj, 2017). Regarding tidal



influence, Dutta et al. (2019b) showed lower pH values during low tide in the Saptamukhi
estuary.

### *DIC and $\delta^{13}C_{DIC}$ variability*

BB estuaries had higher DIC in surface water (862 – 4166 μM; peak in the Veller)
compared to AS (280 – 837 μM) during the wet season. However, in the AS high DIC values
have been reported for the following rivers that feed these estuaries: Narmada (2240 μM);
Tapti (3484 μM); Sabarmati (1760 μM); and Mahisagar (1899 μM) (Sarma et al., 2012; Rao
et al., 2016; Dutta et al., 2015, 2019a, 2021; Samanta et al., 2015; Akhand et al., 2016;
Ganguly et al., 2011; Pattanaik et al., 2017; Piplode and Barde, 2015; Sangani and Manoj,
2017; Bhavya et al., 2017; Fig. 3A & 3B). On average, wet season DIC for the BB estuaries
is ~47% higher than AS. For the dry season, the BB estuaries had comparatively higher DIC
(1541 – 2954 μM; peak in the Hooghly) than AS (Kochi backwater = 1192 μM; Akhand et
al., 2016; Dutta et al., 2019a, 2021; Gupta et al., 2009; Fig. 3A & 3B). The reported DIC
values for the major Indian estuaries are relatively higher compared to other world rivers and
estuaries (Table 2).
During the wet season, the AS estuaries showed wider variability of $\delta^{13}C_{DIC}$ (-5.10 to
-13.00‰; mean: -8.25 ± 2.70‰; peak at the Zuari and Bharatakulza) compared to BB (-2.14
to -7.90‰; mean: -4.18 ± 1.85‰; peak at Ponnayaar). During the dry season, $\delta^{13}C_{DIC}$ varied
between -5.07 and -3.24‰ (mean: -3.78 ± 0.86‰) in the BB estuaries with peak values in the
Matla estuary (Fig. 3C & 3D; Dutta et al., 2019a, 2021; Krishna et al., 2019).

### *Distribution of DOC and POC*

Mean surface water DOC concentration in the BB estuaries (239 – 1079 μM; mean:
418 ± 217 μM; peak in the Ambalayaar) was ~14% higher compared to the AS (37 – 716
μM, mean: 359 ± 172 μM; peak in the Tapti) (Krishna et al., 2015; Ganguly et al., 2011;
Bouillon et al., 2003; Fig. 3E & 3F) during the wet season, while for the BB, DOC varied
between 169 and 497 μM (mean: 322 ± 111 μM) with peak values in the Hooghly estuary
(Dutta et al., 2019a, 2021; Fig. 3E). DOC values reported for the major Indian estuaries were
generally higher compared to those reported for other estuaries worldwide (Table 3).





The mean POC concentration for BB estuaries (51 – 480 µM; mean: 211 ± 142 µM;
peak values in the Godavari River) were ~52% lower compared to the AS (68 – 750 µM,
mean: 321 ± 245 µM; peak values in the Narmada River) (Sarma et al., 2014; Rao et al.,
2016; Fig. 4A & 4B) during the wet season. However, the BB estuaries had ~45% higher
POC (54 – 289 µM, mean: 117 ± 68 µM; peak values in the Hooghly estuary) than the AS
(45 – 98 µM, mean: 64 ± 19 µM; peak values in the Bharatakulza) (Rao et al., 2016; Dutta et
al., 2019a, 2021; Fig. 4A & 4B), during the dry season.
$\delta^{13}C_{POC}$ during the wet season varied between -30.40 and -23.40‰ (mean: -26.36 ±
2.41‰) for the BB estuaries with peak values observed in the Hooghly (Sarma et al., 2014;
Dutta et al., 2019a, 2021; Ray et al., 2015, 2018; Fig. 4C). On average, the AS estuaries had
~1.33‰ lower $\delta^{13}C_{POC}$ values (-31.40 to -22.60‰; mean: -27.68 ± 3.02‰; peak in the
Narmada; Fig. 4D) (Sarma et al., 2014). Dry season $\delta^{13}C_{POC}$ values for the BB varies between
-23.96 and -23.38‰ (mean: -26.36 ± 2.41‰) with peak values in the Saptamukhi estuary
(Dutta et al., 2019a, 2021; Ray et al., 2015, 2018; Fig. 4C).
***Distribution of $CO_2$ and $CH_4$***
$pCO_2$ during the wet season varied over a wide scale (BB estuaries = 248 - 15210
µatm; peak in the Godavari; AS estuaries = 37 - 716 µatm; peak values in the Tapti)
compared to the dry season (BB estuaries = 355-1648 µatm) (Sarma et al., 2012; Dutta et al.,
2019, 2021; Ganguly et al., 2011; Bouillon et al., 2003; Fig. 5A & 5B). On average, wet
season $pCO_2$ for the BB estuaries was ~6 times higher than the dry season. $pCO_2$ values for
the Indian major rivers are higher than those reported for other rivers worldwide (see Table
2). During the wet season, $CH_4$ concentrations in the BB and AS estuaries varied between 4
and 130 (mean: 32 ± 34 nM; peak values in the Ambalayaar river), 5 and 573 (mean: 176 ±
240 nM; peak values in the Netravathi River), respectively (Rao et al., 2016; Dutta et al.,
2015, 2021; Fig. 6A & 6B). In the dry season, $CH_4$ concentrations in the BB and AS estuaries
varied between 5 and 179 nM (mean: 44 ± 47 nM; peak values in the Vaigai River), 18 and
488 nM (mean: 100 ± 137 nM; peak values in the Tapti River), respectively (Rao et al., 2016;
Dutta et al., 2015, 2021; Fig. 6A & 6B). On an average, the AS estuaries had ~5.5 and ~2.3
times higher $CH_4$ concentrations than the BB during the wet and dry seasons, respectively.





The observed range in CH$_4$ concentrations in Indian estuaries is mostly higher compared to
that reported for most of the world's estuaries (Table 4).

## Discussion

The Indian estuaries, where bi-carbonate is the dominant form of DIC (Dutta et al., 2019a),
are oxic in nature and complete to partial DO undersaturation while transiting from the AS
estuaries to the BB estuaries. The aerobic environment indicates the Indian major estuaries as
hotspots for aerobic degradation of organic matter. Concurrently, in the vegetated coastal
wetland, oxygen depleted porewater discharge from intertidal sediment to adjoining estuary
results to low %DO during low tide (Dutta et al., 2015) when higher organic matter
respiration adds H$^+$ to the estuary decreasing pH (Dutta et al., 2019b).Other work has shown
a flux of sediment porewaters to the estuary "proper" during low tide (Dutta et al., 2013,

2017).

### Sources, sinks, and drivers of DIC cycling

Estuarine DIC concentration and speciation is controlled by a variety of mechanisms

including carbonate dissolution/precipitation, community metabolism, and air-water CO$_2$
exchange. Additionally, mixing, surface run-off, groundwater discharge, tidal characteristics
(for vegetated wetlands), anthropogenic discharges, weathering of rocks, and climatic
condition also influence the estuarine DIC pool. These mechanisms are discussed below in
the context of observations made in Indian estuaries.

#### *Chemical weathering, precipitation and physiography of Indian river basins*

Carbonate mineral weathering has been shown to be an important contributor to the

DIC pool of Indian estuaries based on observed δ$^{13}$C$_{DIC}$ – TAlk relationships (significantly
positive; r$^2$ = 0.52, p <0.01; Krishna et al., 2019). Despite higher chemical weathering in the
Deccan Trap basalts (Das et al., 2005; Singh et al., 2005) that occupied the catchments of
north western rivers and upper reaches of the Godavari and Krishna, a larger DIC is reported
in rivers draining over metamorphic rock landscapes. Additionally, despite higher weathering
rates caused by heavy precipitation in the SW region of the Indian sub-continent (Gupta et





al., 2011), lower DIC concentrations are reported there. This suggests alternate drivers of
DIC behavior in this region, as discussed below.

Krishna et al. (2019) proposed the degree of precipitation as the major cause of low

estuarine DIC levels based on the exponential decrease in DIC with precipitation ($r^2 = 0.72$).
Our individual analysis of BB and AS datasets shows a significant relationship existing
between wet season DIC and precipitation with linear and exponential relationships,
respectively, for the two marginal seas (Fig. S1). This suggests that the variability of
precipitation plays a an important, but varying role in controlling DIC in both BB and AS
estuaries. DIC has also been shown to be positively correlated with the length of the rivers ($r^2$
$= 0.38$, $p < 0.01$; Krishna et al., 2019). Riverine DIC has been reported to increase along the
course of the fluvial network (Hotchkiss et al., 2015) due to an increase in the residence time
of water (Catalan et al., 2016). The comparatively smaller rivers draining into AS estuaries
reduces the residence time of water, with less opportunity for organic matter to be
remineralized to DIC (Krishna et al., 2019).
***Estuarine mixing***

DIC and $\delta^{13}C_{DIC}$ values generally increase linearly with increasing salinity in the

Indian estuaries during both wet and dry periods (Fig. S2). However, the DIC – salinity
relationship for BB estuaries fits well with a polynomial relationship for the wet season DIC
at salinities >12 (Fig. S2A). These statistical analyses indicate that the degree of marine and
fresh waters mixing plays a crucial role in regulating DIC budgets of Indian estuaries. The
same was previously reported for the Hooghly estuary (Dutta et al., 2019a, 2021; Samanta et
al., 2015) as well as Godavari estuary (Bouillon et al., 2003) from the Indian sub-continent.
These studies applied a two-end member mixing model for these estuaries to quantitatively
understand processes links with DIC addition/removal. Here, the proportional relationship
between salinity and $\delta^{13}C_{DIC}$ is well explained based on the fact that $\delta^{13}C_{DIC}$ of marine water
is greater than the $\delta^{13}C_{DIC}$ of freshwater. However, the proportional relationship between
salinity and DIC despite the concentration of DIC of marine water being less than the
concentration of DIC freshwater (Sabine et al., 2002; Sarma et al., 2012) suggests additional
DIC inputs to Indian estuaries via other pathways discussed below.





### *Groundwater DIC discharge*


Groundwater plays a pivotal role in regulating elemental concentrations as well as

isotopic signatures of rivers and estuaries if they are fed by aquifers (Samanta et al., 2015).
At the mouth of the Ganga–Brahmaputra system in Bangladesh, Moore (1997) reported the
role of submarine groundwater discharge (SGD) on controlling the abundance and
distribution of selected elements (e.g., Ba) and isotopes (e.g., $^{226}$Ra). There are relatively few
studies of SGD in Indian estuaries, but several recent datasets on groundwater DIC exist from
the Indo-Gangetic basin. Previously, Samanta et al. (2015) reported a wide variability in
shallow groundwater DIC concentrations (4.39 – 11.21 mM) and $\delta^{13}C_{DIC}$ (-13.3‰ to -2.3‰)
from the surrounding regions of the Hooghly estuary. Dutta et al. (2019a) reported a similar
range of values during a post-monsoonal study on Hooghly-Sundarbans systems (Hooghly:
DIC = 5.66 – 11.76 mM, $\delta^{13}C_{DIC}$ = -12.66‰ to -6.67‰; Sundarbans: DIC = 7.52 – 13.60
mM; $\delta^{13}C_{DIC}$ = -18.05‰ to -6.84‰) covering the entire stretch starting from freshwater to
marine regimes. In both cases, groundwater DIC concentrations were greater than surface
water concentrations, suggesting that SGD is an important source of DIC to the Indian
estuaries. Mixing calculations performed for the low salinity region of the Hooghly estuary
suggest that SGD contributes to ~5 – 20% of the estuarine DIC pool, though these
calculations were based on Ca and salinity, not direct DIC flux measurements (Samanta et al.,
2015). Contrasting these findings, Somayajulu et al. (2002) found limited evidence for
groundwater contribution in the Hooghly estuary based on 'radium' isotopes.

For vegetated coastal wetlands (e.g., mangroves, seagrass, and saltmarsh), exchange

fluxes between sediment porewaters and estuarine surface waters play a significant role in
regulating DIC budgets (Maher et al., 2013, 2016; Dutta et al., 2015; Tait et al., 2016;
Bouillon et al., 2007). Dutta et al. (2019a) estimated porewater DIC levels in the Indian
Sundarbans mangrove system to be 13.43 mM (~6 times higher than surface water DIC) with
depleted $\delta^{13}C_{DIC}$ signatures (-18.05‰). The reported porewater DIC concentration in this
mangrove system is much higher than other mangroves around the world (Bouillon et al.,
2007; Taillardat et al., 2018; Maher et al., 2013). The porewater – surface water DIC
exchange flux was estimated to be 770 mmol m$^{-2}$ d$^{-1}$ based on the DIC concentration



gradient, porewater specific discharge, and porosity (Dutta et al., 2019a). Integrating this flux
over the entire intertidal zone of the Indian Sundarbans mangroves (45% of total forest area;
http:// www.sundarbanbiosphere.org/html_files/sunderban_biosphere_reserve.htm), total DIC
export from intertidal mangrove sediments to the estuary is estimated to be ~6.37 Tg C yr$^{-1}$.
Furthermore, Ray et al. (2018) estimated a DIC export from the estuaries of the Indian
Sundarbans to the adjoining BB of ~3.69 Tg C yr$^{-1}$. Considering very limited anthropogenic
inputs to the estuaries of the Sundarbans (Dutta et al., 2015), this calculation suggests that
~58% of total DIC export from the Sundarbans mangrove sediment is transported to the BB
and the rest either increases estuarine DIC pools or is removed within the estuary via
biogeochemical processes. DIC removal in the estuaries of the Indian Sundarbans is also
evident during the post-monsoonal period when stable isotopic signatures suggest a large DIC
output compared to input via mangrove-derived OC mineralization (Dutta et al., 2019a)
*Anthropogenic DIC discharge*

Although anthropogenic C fluxes are not reported for most of the Indian estuaries, the

relationship between population density and DIC can be used as a proxy to examine
anthropogenic influences (Krishna et al., 2019). Krishna et al. (2019) proposed significant
anthropogenic contributions to estuarine DIC based on the linear relationship between
population density and DIC ($r^2$ = 0.41, p <0.01 excluding the Sabarmati and Mahisagar
estuaries). However, our data analysis separating BB and AS estuaries shows no significant
relationship during the wet season (Fig. S3), suggesting limited anthropogenic influence on
DIC. Our findings are supported by pre-monsoon measurements in the anthropogenically
stressed Hooghly estuary (Dutta et al., 2021); although population density data was not
available for this study region, both sides of the river bank are occupied by the very densely
populated city including the megacity Kolkata as well as several jute and other industries that
supplies 1154 million L$^{-1}$ of anthropogenic discharge on a daily basis (Dutta et al., 2019a;
Ghosh, 1973; Khan, 1995). Despite these large anthropogenic discharges, the study identified
a predominance of estuarine algae and marine plankton in the POC pool of the Hooghly
estuary and from that they proposed the anthropogenic organic C (i) either triggered
productivity (but no evidence for increased productivity was observed), (ii) principally exists
in the DOC phase, or (iii) if it exists principally as POC, its biogeochemical modification is
happening in the particulate phase. These uncertainties highlight the need for detailed
quantification of anthropogenic DIC fluxes to the Indian estuaries considering the widespread
and ever-expanding human development and activities along the Indian coastline over the
years.
***Hydrological and biogeochemical drivers of DIC cycling***

During the wet season, a significant positive relationship exists between $\delta^{13}C_{DIC}$ –

DIC for the both BB and AS estuaries (Fig. S4). However, the relationship turns negative for
the dry season in BB estuaries (Fig. S4A). The positive relationship during the wet season
suggests that $^{13}C$ enriched DIC is exported to the estuaries, which is perhaps related to
carbonate dissolution. Supporting this hypothesis, Samanta et al. (2015) showed calcite
saturation index values less than 0 (i.e., calcite dissolution) for all monsoonal samples
collected from the high saline region (salinity ≥ 10) of the Hooghly estuary. The negative
relationship for BB estuaries during the dry season is perhaps caused by OM mineralization,
as evidenced primarily by Bouillon et al. (2003) for the Godavari estuary and very recently
by Dutta et al. (2021) for the Hooghly estuary during their pre-monsoonal surveys.
Coinciding with this argument, Dutta et al, (2015) together with earlier studies by
Mukhopadhyay et al. (2006), Biswas et al. (2007) showed that the Hooghly-Sundarbans
system is net heterotrophic during the dry season (i.e., community respiration: productivity
>1). Furthermore, based on the calculated $^{13}C$ value of respired C (Godavari = −28.6‰;
Hooghly = −12‰) the authors proposed the potential role of estuarine algae ($\delta^{13}C$: − 12 to −
23‰; Smith and Epstein, 1971) and mangroves ($\delta^{13}C$: − 27‰; Miyajima et al., 2009) for DIC
addition by respiration in the Hooghly and Godavari estuaries, respectively.

Despite a lack of $\delta^{13}C_{DIC}$ data unavailability for many of the Indian major estuaries,

indirect relationships between different parameters as well as existing community metabolism
data for some estuaries may also highlight the biological influence on DIC. AS estuaries with
elevated phytoplankton levels (i.e., Chl *a* > 5 mg m$^{-3}$), Krishna et al. (2019) showed an
indirect signature of phytoplankton productivity based on the negative DIC – Chl *a*
relationship ($r^2$ = 0.44, p <0.01). The same was again confirmed by the positive relationship



between $\delta^{13}C_{DIC}$ and Chl *a* ($r^2$ = 0.49, p <0.01) considering preferential $^{12}C$ uptake over $^{13}C$
leaves the residual DIC enriched in $^{13}C$ as during photosynthesis. Additionally, based on
gross primary productivity and community respiration estimates by oxygen monitoring in
light/dark bottles, Gupta et al. (2009) showed that the Cochin estuary was net autotrophic
throughout the seasons (i.e., net DIC removal). However, contrasting conditions have been
observed for the Mahanadi, Mandovi and Zuari estuaries. Pattanaik et al. (2019) estimated
that the Mahanadi estuary was predominantly net autotrophic, whereas Ganguly et al. (2011)
showed the same system to be net heterotrophic during the monsoon but fluctuated between
net autotrophic and heterotrophic during the transition from pre- to post-monsoon periods.
For the Mandovi and Zuari rivers, Ram et al. (2003) showed a transition from net autotrophy
during the non-monsoon seasons to net heterotrophy during the monsoon season by the
application of $^{14}C$ assimilation methods.

In addition to the aforementioned proxies, $n$DIC – $n$TAlk relationships have been

used to identify the active biogeochemical processes in the surrounding estuaries near
vegetated coastal wetlands. Previously, Dutta et al. (2019b) and thereafter Akhand et al.
(2021) proposed the potential impact of denitrification, sulphate reduction and organic matter
respiration in controlling DIC in the estuaries of the Sundarbans region based on the
significant relationship between $n$DIC and $n$TAlk (Dutta et al., 2019b: $r^2$ = 0.43, p <0.05,
slope = 0.89). In addition, using the same approach, Borges et al. (2003) showed sulphate
reduction together with organic matter respiration controlled DIC while investigating $CO_2$
dynamics in the mangrove-dominated Gaderu creek, India ($r^2$ = 0.945, slope: 0.61 ± 0.03).
Indian mangrove systems appear to behave similar to Australian and Vietnam mangrove
settings (Sippo et al., 2016). Considering the studied estuaries are all generally oxygenated,
the anaerobic signals (as mentioned earlier) might be derived from the intertidal mangrove
sediments during porewater exchange as proposed by Dutta et al. (2019b). This needs to be
further examined for Indian coastal systems considering their diverse nature.
**Sources, sinks, and drivers of DOC cycling**

Estuarine DOC pools include both allochthonous and autochthonous origin (Ward et

al., 2017). The major sources of allochthonous DOC are leaching of terrestrial OM (present



in soils, debris of terrestrial plants, wood, and leaf litter) in the catchment area as well as
localized inputs via anthropogenic discharges, which consists of both domestic and industrial
sewages (Bin and Longjun, 2011; Ray et al., 2018; Dutta et al., 2019a). Precipitation (Sarma
et al., 2014) together with tidal flushing (for vegetated wetlands) carry terrestrial DOC to
rivers and subsequently to their estuaries (Dutta et al., 2019a, Maher et al., 2013).
Autochthonous DOC sources include phytoplankton, autolysis of bacteria, bacteria and
macrophytes, viral lysis of bacteria and phytoplankton, zooplankton grazing and excretion
(Carlson et al., 1994; Bianchi et al., 2004; Bronk et al., 1994; Diaz and Raimbault, 2000;
Fuhrman, 1999; Wilhelm and Suttle, 1999; Middelboe and Jorgensen, 2006; Berman and
Bronk, 2003). Transformation of DOC through physical (e.g., flocculation and
sorption/desorption), photochemical, and biological processes alter the signature of these
DOC sources as they are transported through estuaries to the continental shelf (Ray et al.,
2018, Dutta et al., 2019a, 2019b).
***Terrestrial DOC fluxes***

Terrestrial DOC fluxes normalized to catchment area (i.e., DOC yields) can vary by

orders of magnitude. Krishna et al. (2015) calculated catchment area normalized fluxes of
DOC to Indian estuaries during the dry season and it accounts for 35 to 1903 g C m$^{-2}$ yr$^{-1}$.
This is comparable to fluxes estimated for rivers around the world, which vary by an
additional order of magnitude (0.1 – 5695 g C m$^{-2}$ yr$^{-1}$; Alvarez-Cobelas et al., 2012).

For Indian estuaries, there was no significant relationship between DOC fluxes with

freshwater discharge (r$^2$ = 0.01, p = 0.60) and catchment area of the river (r$^2$ = 0.05, p =
0.30), suggesting that these factors may not be the dominant control of terrestrial DOC fluxes
in the region (Krishna et al., 2015). When we re-analysed BB and AS estuarine (Fig. 12A)
data separately, catchment area and DOC fluxes remained uncorrelated (BB estuaries: r$^2$ =
0.17, p = 0.16; AS estuaries: r$^2$ = 0.18, p = 0.20; Fig. not shown). The earlier report together
with our data analysis suggests variability of catchment area is not a major governing factor
for DOC. However, DOC yield was strongly correlated with rainfall (r$^2$ = 0.87, p = 0.06), soil
organic carbon content (r$^2$ = 0.94, p = 0.02), and biomass carbon (r$^2$ = 0.95, p = 0.02)
(Krishna et al., 2015). Additionally, higher DOC fluxes were estimated for AS estuaries,



which may be the result of intense DOC scrubbing from OC-rich soils by heavy rainfall
during the SW monsoon (~3000 mm) (Soman and Kumar, 1990; Kishwan et al., 2009).
***Groundwater DOC discharge***
To our knowledge, no data is available for assessing the contribution of groundwater
discharge to the DOC pools of Indian estuaries. Additionally, vegetated ecosystems along the
coast add DOC to the adjoining estuaries through pore-water exchange, but no direct data is
available on porewater mediated DOC export. However, indirect signatures of these fluxes
have been observed. In the Pichavaram mangroves, SE coast of India, Ranjan et al. (2010)
reported porewater DOC concentrations of 2071μM, which is higher than surface water
values (166 – 1954 μM). The concentration gradient suggests that DOC export via porewater
exchange more than likely occurs, but unfortunately a lack of other associated hydrological
parameters needed to compute lateral exchange restricts us from calculating fluxes.
Additionally, a diurnal study in the Indian Sundarbans by Dutta et al. (2019b) hypothesized
that porewater mediated DOC exchange was the driver of ~ 30 μM higher average DOC
concentrations during low tide compared to high tide.
***Anthropogenic DOC discharge***
The population density-DOC relationship shows distinct characteristics for the BB
estuaries under population levels $< 300$ per $km^2$ and $>300$ per $km^2$. Under $< 300$ per $km^2$ the
DOC – population density relationship showed a significant positive correlation (Fig. S5A).
However, no significant relationship exists for population densities $>300$ per $km^2$ nor for the
AS estuaries across the entire range of population densities (Fig. S5). This suggests limited
anthropogenic impact on DOC in the Indian estuaries with the exception of BB systems with
population densities less than 300 per $km^2$ (that includes Mahanadi, Vamsadhara, Nagavali,
Godavari, Krishna, Penna, Ponnayaar estuaries). It is possible that anthropogenic inputs
primarily influence POC pools, which is evident from $\delta^{13}C_{POC}$ values (see Fig. 4C & 4D). To
properly understand the magnitude and impact of anthropogenic DOC inputs to Indian
estuaries, more thorough investigations on $\delta^{13}C_{DOC}$ and other organic tracers of
anthropogenic activity are needed.
***Transformations driving non-conservative behaviour of estuarine DOC***





DOC generally behaves non-conservatively in Indian estuaries as evident from non-
linear DOC – salinity relationships (Fig. S6A & S6B). These non-conservative behaviours
have been previously reported by the Dutta et al. (2019a, 2019b, 2021) for the Hooghly-
Sundarbans estuarine systems based on inter-spatial and diurnal variabilities. Krishna et al.
(2015) showed no potential contribution of autochthonous DOC during the monsoon period
based on its relationship with Chl $a$ ($r^2$ = 0.004, p = 0.77). However, our dry season BB data
analysis shows a significant link between DOC and Chl $a$ (Fig. S6C) with an exponentially
decreasing trend. This link suggests that unlike during the wet season, autochthonous DOC is
an important source during the dry season DOC. The decreasing trend might be a signal of its
simultaneous removal from the system considering algal-derived DOC is generally labile and
may even promote priming effects that further degrade terrestrial DOC sources (Bianchi et
al., 2015; Ward et al., 2016; 2019).
The mean DOC/DON ratio (8.4 ± 3.8) for Indian estuaries as calculated by Krishna et
al. (2015) is close to the mean POC/PON ratio (8.7 ± 2) calculated by Sarma et al., (2014)
and the biologically available DOC fraction in the global coastal ocean (8.8 ± 4.4) (Lonborg
and Alvarez-Salgado, 2012). However, it is lower than that reported for the continental
margins of the global oceans (DOC/DON = 17.8) (Lonborg and Alvarez- Salgado, 2012) and
terrestrial refractory DOM (DOC/DON = 29.6) (Meybeck, 1982). Based on these ratios,
Krishna et al. (2015) proposed that the DOC pool in Indian estuaries is primarily composed
of high-quality non-refractory DOC.
The POC/DOC ratio can be used as a proxy to understand the impact of POC on DOC
cycling. Based on the reported dataset, our calculated POC/DOC data for the BB estuaries
(wet season: 0.55 ± 0.40; dry season: 0.36 ± 0.14) is relatively lower compared to the AS
(0.82 ± 0.79 except for the Netravathi having a very high value of 10.8). Based on the
differences between the two regions, we propose that POC-DOC conversion might be more
active in the BB estuaries. However, this is the case for the BB only during the dry season
when DOC increases with increasing POC (Fig. S7A). The opposite condition occurs during
the wet season (Fig. S7A) and for the AS (Fig. S7B). This observation is similar to pre-
monsoon spatial surveys in the Hooghly estuary (Dutta et al., 2021). Based on the DOC –



POC relationship it was proposed that DOC removal via POC formation was more efficient
than DOC formation via POC dissolution.

Regarding DOC photo-decomposition, no direct experiments have been conducted in

Indian estuaries to our knowledge. However, diel measurements of day/night DOC variability
suggest that photo-oxidation may have a limited influence on DOC levels in the Indian
Sundarbans (Dutta et al., 2019b). It was hypothesized that unstable water conditions
(Richardson number <0.14) leading to intensive vertical mixing with longitudinal dispersion
coefficients of 784 $m^2$ $s^{-1}$ limited the potential for photo-decomposition to occur (Sadhuram
et al., 2005; Goutam et al., 2015).

Biological mineralization of DOC to $CO_2$ while transiting through the coastal ocean is

another important pathway of DOC removal (Sarma et al., 2012; Dutta et al., 2019a). The
DOC – $pCO_2$ relationship is not significant for the BB estuaries (Fig. S7C). For the wet
season in AS estuaries, the nature of the DOC – $pCO_2$ relationship is different for $pCO_2$ <
6800 µatm and $pCO_2$ > 6800 µatm conditions. When $pCO_2$ is less than 6800 µatm, there is a
significant positive relationship between DOC and $pCO_2$ in contrast to conditions when $pCO_2$
is greater than 6800 µatm and DOC shows a significant negative relationship with $pCO_2$ (Fig.
S7D). The non-significant relationship for the BB during the wet season suggests that either
there are limited DOC mineralization rates, or other key drivers of $pCO_2$ during this time.
Dutta et al. (2019a) reported the same for the Hooghly estuary during the post-monsoon
season. For the AS, a positive relationship between DOC and $pCO_2$ when $pCO_2$ is less than
6800 µatm suggests that DOC mineralization may be an important source of $CO_2$ to the
system. However, the significant negative relationship under $pCO_2$ > 6800 µatm conditions
suggests a decrease of aerobic bacterial activity with increasing DOC. In this case, another
possibility is potential DOC mineralization and simultaneous removal of $CO_2$ by $CO_2$
outgassing (discussed later), primary productivity, carbonate precipitation, and/or export to
the adjoining continental shelf.
**Sources, sinks, and drivers of POC cycling**
As with DOC, estuarine POC pools include both autochthonous and allochthorounous POC.
Depending upon environmental conditions, the mixing between marine and fresh waters,





inputs via terrestrial ecosystems, in situ biogeochemical processes, and anthropogenic inputs
all contribute to the POC pool and mediate POC transformations.
***Natural and anthropogenic POC sources***

The stable isotopic composition of POC ($\delta^{13}C_{POC}$) is often used to identify sources of

particulate organic matter in estuaries. The utility of this tracer can sometimes be diminished
by high particulate loads and longer water residence times in certain Indian estuaries (Sarma
et al., 2014); nonetheless it is the primary tool that has been used to evaluate POC origins in
Indian tracers and there has been little use of other tools such as organic biomarkers in the
region.

During the wet season, $\delta^{13}C_{POC}$ values across the Indian estuaries show dominant

POC contributions from freshwater algae for both BB and AS estuaries, $C_3$ plant material for
BB estuaries, and marine organic matter for AS estuaries (Fig. 4C & 4D). However,
anthropogenic inputs are also evident during the dry season in the Hooghly estuary and the
estuaries of the Sundarbans. Despite a wide range of cultivation of $C_4$ plants (e.g., Ragi, Bajra
and Jowar) and $C_3$ plants (mostly wheat and rice) along the coast of the BB and AS,
respectively, estuarine $\delta^{13}C_{POC}$ signatures are substantially different than $\delta^{13}C$ of these
terrestrial plants. Regarding sewage contributions, the megacity Kolkata and some other
highly populated cities (e.g., Howrah, North and South 24 Parganas) supply a large amount of
municipal and domestic waste to the Hooghly estuary on a daily basis. The estuaries of Indian
Sundarbans, on the other hand, have very limited anthropogenic discharges that mostly only
occur during the monsoon (Dutta et al., 2015); the signature of these discharges can outweigh
isotopic signatures of mangrove vegetation ($\delta^{13}C \sim$ -27‰; Miyajima et al., 2009) during this
period. Population density and POC relationships are not significant for the BB or AS
estuaries (Fig.S8), suggesting limited anthropogenic POC inputs to Indian estuaries.
However, $\delta^{13}C_{POC}$ data clearly suggests anthropogenic POC contributions, especially for the
BB (Fig. 4C & 4D). The contrasting findings between bulk and isotopic observations
demands a comprehensive investigation on anthropogenic POC inputs to Indian estuaries,
perhaps leveraging molecular biomarkers.
***Biogeochemical drivers of POC cycling***





The relationships between POC and $\delta^{13}C_{POC}$ with salinity in Indian estuaries are not
significant (Fig.S9). This suggests that freshwater mixing is not the major driver of POC
composition or concentrations. Regarding aerobic mineralization, the relationship between
$\delta^{13}C_{POC}$ and %DO are also not significant for the BB estuaries during both wet and dry
seasons (Fig. S10A). In contrast to the BB, there was a significant negative relationship for
the AS estuaries during the wet season (Fig. S10B). Our statistical analysis suggests that
variability of %DO does not play an important role in POC transformations for the BB
estuaries; however, contrasting reports exist regarding POC respiration in the Hooghly-
Sundarbans system. During a post-monsoon survey, Dutta et al. (2019a) observed POC
mineralization in freshwater regions of the Hooghly estuary as well as Sundarbans. But
similar to these statistical analyses, a recent pre-monsoon study by Dutta et al. (2021)
reported limited POC respiration in the Hooghly-Sundarbans systems. In contrast to the BB
estuaries, the significant negative relationship in the AS suggests that aerobic POC
mineralization plays an important role in transforming POC, which was also proposed by
Sarma et al. (2012) when examining all Indian estuaries together. Our data analysis
separating BB and AS datasets predicts only active POC respiration for the AS, which is also
evident in $p$CO$_2$ trends.
Despite primarily oxygenated conditions in the surface waters of Indian estuaries, it is
possible that anaerobic processes also transform and/or decompose POC, perhaps related to
sediment transport and resuspension dynamics. For BB estuaries, there is a significant linkage
between $\delta^{13}C_{POC}$ and CH$_4$ during both wet and dry seasons (Fig.S11A). The relationship
might suggest CH$_4$ production via anaerobic POC degradation (methanogenesis), which was
reported by Dutta et al. (2021) for the Indian Sundarbans. In contrast to the BB, there is an
exponential relationship between $\delta^{13}C_{POC}$ and CH$_4$ in the AS, which may suggest some
linkage between estuarine POC and CH$_4$ cycling dynamics (Fig. S11B).
**Sources, sinks, and drivers of CO$_2$ cycling**
Estuarine $p$CO$_2$ is principally controlled by community metabolism (i.e., balance
between respiration and primary production) as well as carbonate precipitation and





dissolution. In addition, hydrological (e.g., estuarine mixing and groundwater discharge) and
physical (air-water $CO_2$ exchange) processes also control the level of variability of $pCO_2$.
*Riverine $CO_2$ sources*
Taking all Indian estuarine data together, Sarma et al. (2012) showed a significant
positive relationship between wet season $pCO_2$ and river discharge ($r^2 = 0.71$; $p < 0.001$
excluding the largely anthropogenically stressed Tapti estuary). But our data analysis shows
contrasting result when BB and AS estuaries are analysed separately. Excluding the
Ponnayaar, the wet season $pCO_2$ - discharge relationship is significant and positive for the
BB estuaries ($r^2 = 0.82$, $p < 0.001$; Fig.S12A). Low river discharge favours a higher
proportion of marine water within the estuary, resulting in low $pCO_2$. However, there was no
significant relationship between $pCO_2$ and discharge for AS estuaries (Fig. S12B).
*Groundwater $CO_2$ sources*
To our knowledge, no data exists to evaluate fresh groundwater contributions to
estuarine $pCO_2$ for Indian estuaries. However, Akhand et al. (2021) reported porewater $pCO_2$
values up to 5423 µatm in the Indian Sundarbans mangroves. Using mean annual soil
temperature and porewater salinity from Dutta et al., (2013), we estimate that this equates to a
$CO_2$ concentration of 137 µM. By using porewater-specific discharge and porosity (Dutta et
al., 2013, 2015), it is estimated that $CO_2$ export by porewater exchange with the adjoining
river is ~ 7.89 mmol m$^{-2}$ d$^{-1}$. This value is much lower compared to that reported for the
North creek, New South Wales, Australia (1622 mmol m$^{-2}$ d$^{-1}$; Atkins et al., 2013).
Extrapolating the flux over the entire intertidal area of the Indian Sundarbans mangrove
system, total $CO_2$ export flux via pore-water is calculated as 0.24Tg C yr$^{-1}$, which is ~3.8% of
the total DIC export. This calculation suggests that pore-water DIC principally includes
carbonate and bi-carbonate rather than $CO_2$.
*Anthropogenic $CO_2$ sources*
For the BB, $pCO_2$ shows a significant negative relationship with population density
(Fig. S13A). However, the relationship is not significant for the AS estuaries (Fig. S13B).
This analysis suggests that anthropogenic $CO_2$ inputs might impact $pCO_2$ in BB estuaries but
not in the AS. The lack of a significant relationship with DIC (discussed earlier) together with



$p$CO$_2$ decreasing with increasing population in the BB might be an indicator of removal of
$p$CO$_2$ driven by anthropogenic inputs; for example, nutrient inputs may promote increased
primary productivity and/or eutrophication.
***Biogeochemical drivers of CO$_2$ cycling***

During the wet season, $p$CO$_2$ shows a significant negative relationship with %DO in

both the BB and AS estuaries (Fig.S14). The significant relationships suggest occurrence of a
mechanism that produces CO$_2$ with simultaneous consumption of dissolved O$_2$ within the
water column, i.e., organic matter mineralization. This relationship also holds up when
analysing all Indian estuarine data together (Sarma et al., 2012; $r^2 = 0.56$, p $< 0.001$). Sarma
et al., (2012) confirmed that organic matter mineralization drove this relationship based on
the positive [CO$_2$*]$_{Excess}$ – apparent O$_2$ utilization (AOU) relationship. AOU calculations
were not possible for our compiled dataset of BB and AS estuaries, so we extracted data from
Sarma et al. (2012) using a graph reading tool (http://www.graphreader.com/). Excluding two
data points having maximum [CO$_2$*]$_{Excess}$ and minimum AOU, respectively (marked in Fig.
21), the [CO$_2$*]$_{Excess}$ and AOU slope for the major Indian estuaries is calculated as 2.43. This
is much higher than the theoretical value for Redfield respiration ($\Delta$CO$_2$/-$\Delta$O$_2$ = 0.90; Zhai et
al., 2005), suggesting higher wet season CO$_2$ production in the Indian estuaries than expected
from Redfield respiration. However, the reverse case applies for the BB during the dry season
when the $p$CO$_2$ - %DO relationship is not significant ($r^2 = 0.34$, p = 0.22; Fig.S14A),
indicating limited impact of organic matter respiration on $p$CO$_2$. In this regard, during a post-
monsoonal survey Dutta et al. (2019a) showed that organic matter respiration played a
significant role in CO$_2$ production in the estuaries of Sundarbans but not in the Hooghly
estuary. An opposite trend was reported during pre-monsoon season (Dutta et al., 2021).

Other than aerobic respiration, nitrification also plays a crucial role in increasing

estuarine pH, which in turn favours greater CO$_2$ outgassing to the atmosphere (Billen, 1975,
Frankignoulle et al., 1996). In these oxygenated estuaries, Sarma et al. (2012) showed higher
NH$_4^+$ in the west coast rivers (1.4-16.6 mM) than the east coast rivers (0.2-7.0 mM).
Although Miranda et al. (2008) hypothesized that it is unlikely that nitrification could be an
important mechanism for mitigating NH$_4^+$ pollution in the Kochi Backwaters (drains into the





AS), they estimated nitrification rates between 0.06 and 166 nmol N L$^{-1}$ hr$^{-1}$ there.
Additionally, a recent study by Dutta et al. (2019b) also revealed that nitrification played an
important role in the estuaries of the Indian Sundarbans based on the very high diurnal $\delta^{15}N_{PN}$
(8.71–14.75‰) compared to other Indian estuaries (northern rivers: 0.7 - 5.8‰, southern
rivers: 5 - 10.3‰; Sarma et al., 2014). Preferential $^{14}N$ uptake during nitrification (Mariotti et
al., 1984) results in $^{15}N$ enriched $NH_4^+$ pool, which in turn results to higher $\delta^{15}N_{PN}$ when
incorporated by algae (Mariotti et al., 1984) and heterotrophic bacteria (Middelburg and
Nieuwenhuize, 2000). The limited amount of work on nitrification in Indian estuaries
suggests that it may play a role in $p$CO$_2$ cycling, but more systematic studies are essential to
fill up the data gap in this topic.
***Air-water CO$_2$ exchange***
The flux of CO$_2$ to the atmosphere ($F$CO$_2$) during the wet season varies between -0.02
to 96.32 and 3.24 to 362.45 mmol m$^{-2}$ d$^{-1}$ for the BB and AS estuaries, respectively (Fig. 5C
& 5D). In contrast, during the dry season $F$CO$_2$ is substantially lower (BB estuaries: -4.67 to
30 mmol m$^{-2}$ d$^{-1}$; AS estuaries: 1.30 – 2.50 mmol m$^{-2}$ d$^{-1}$). Positive and negative values (net
emission and uptake, respectively) for the BB estuaries suggest that the estuaries act as both
CO$_2$ sources and sinks. The AS estuaries, on the other-hand, are persistent CO$_2$ sources to the
atmosphere. The negative $F$CO$_2$ values for BB estuaries are mostly associated with the
Rushikulya during the wet season and major estuaries of the Indian Sundarbans during the
dry season.
**CH$_4$ dynamics in Indian estuaries**
***General sources and sinks of CH$_4$***
In the anoxic environment, CH$_4$ produces in the terminal step of the organic matter
decomposition when all the electron acceptors consume and electron donors are surplus
(Dutta et al., 2017). The produced CH$_4$ enters in the estuaries by lateral transport from the
upstream river and inputs from the sediments via diffusion and groundwater discharge.
However, the removal of CH$_4$ includes aerobic, anaerobic oxidations and outgassing to the





regional atmosphere. In addition to this, stratification of water column also promotes $CH_4$
production (Rao et al., 2016).
***Riverine $CH_4$ sources***
In the Indian estuaries, $CH_4$ - discharge relationships are not significant (Fig. S16) excluding
the AS during the dry season where the relationship is significant (Fig. S16B). This statistical
analysis suggests that freshwater discharge only plays a major role in controlling the
concentration of $CH_4$ in the AS estuaries during the dry season. An inverse relationship
between $CH_4$ and salinity has also been reported for estuaries worldwide (Zhang et al., 2008;
Middelburg et al., 2002; Koné et al., 2010; Bange, 2006; Borges et al. 2015), which is
associated with oxidation and outgassing removing freshwater-derived $CH_4$ along the
estuarine gradient.
***Groundwater $CH_4$ sources***
Groundwater discharge is considered to play a pivotal role in controlling $CH_4$ budgets
in estuaries, particularly in mangrove dominated estuaries (Dutta et al., 2015). Biswas et al.
(2007) reported porewater $CH_4$ concentration of 5769 nM in the Indian Sundarbans. After
almost a decade, in the same ecosystem Dutta et al. (2015) reported porewater $CH_4$
concentrations in intertidal (1881 – 3370 nM) and subtidal (2070 $\pm$ 1039 to 3980 $\pm$ 1227 nM)
sediments, which had significantly higher concentrations than surface waters (54 $\pm$ 5 to 91 $\pm$
21 nM). The concentration gradient results in advective and diffusive $CH_4$ fluxes on the order
of 116 $\pm$ 31 to 199 $\pm$ 48 $\mu$mol m$^{-2}$ d$^{-1}$ and 7 $\pm$ 2 to 10 $\pm$ 2 $\mu$mol m$^{-2}$ d$^{-1}$, respectively.
Extrapolating these fluxes over the entire Indian Sundarbans it was estimated that
groundwater contributed ~1.88 Gg $CH_4$ yr$^{-1}$ to surrounding estuaries, ~99% of which
advective flux via porewater exchange across the intertidal sediment-river interface (Dutta et
al., 2015). Additionally, Rao et al. (2016) reported mean ground water $CH_4$ concentrations for
the Godavari and Krishna estuaries of 1566 $\pm$ 81nM. The same study estimated groundwater -
estuary advective $CH_4$ fluxes during the dry season of 19.2 and 22.4 $\mu$mol m$^{-2}$ d$^{-1}$ in the
Godavari and Krishna rivers, respectively. However, sediment-water $CH_4$ fluxes were
reported as 20.9 $\pm$ 3 and 25.1 $\pm$ 4 $\mu$mol m$^{-2}$ d$^{-1}$ in the Godavari and Krishna rivers,
respectively. The author also proposed that ~40% of the $CH_4$ budget in the Godavari and





Krishna estuaries was driven by the above-mentioned pathways. Groundwater $CH_4$ fluxes
have not been studied in most of the other Indian estuaries, meriting a comprehensive
investigation for future $CH_4$ budgets for Indian estuaries.

### *Anthropogenic $CH_4$ sources*

Wastewater end member $CH_4$ data has not been studied for the major Indian rivers
and estuaries. Alternatively, $CH_4$ – population density relationships can be used as a proxy to
understand the impact of anthropogenic inputs. The relationships showed limited significance
of anthropogenic inputs on $CH_4$ concentrations in the Indian estuaries (Fig.S17) but this
should be confirmed by stable isotopic analysis of $CH_4$ as well as quantification of $CH_4$
concentrations in wastewater inputs.

### *Biogeochemical drivers of $CH_4$ cycling*

The significant link between POC and $CH_4$ in Indian estuaries was previously
discussed. In terms of methane oxidation, the oxygenated waters of Indian estuaries can only
support aerobic $CH_4$ oxidation. Dutta et al. (2015a) reported $CH_4$ oxidation rates in the Indian
Sundarbans ($12.96 \pm 2.86$ to $30.22 \pm 6.46$ nmol $L^{-1} d^{-1}$), but the process might not influence
$CH_4$ distribution significantly except for the AS during the wet season as evident from the
$CH_4$ - %DO relationships (Fig. S18). During aerobic oxidation, $CH_4$ converts to $CO_2$ (Dutta
et al., 2017). In the case of AS during the wet season, the $CH_4$ - $pCO_2$ relationship was
positive and significant; in our other analyses there were not significant relationships except
for a negative relationship between $CH_4$ and $pCO_2$ for BB during the dry season (Fig.S19).
The significantly positive $CH_4$ - $pCO_2$ relationship during the wet season might be linked to
similar sources of $CH_4$ and $pCO_2$ as previously proposed by Borges and Abril (2011).
However, during the dry season, the significant negative $CH_4$ - $pCO_2$ relationship might be
linked with $CH_4$ oxidation (which is not evident from the $CH_4$ - %DO relationship). As
previously mentioned, investigation of the stable isotopic composition of $CH_4$ is needed to
understand how important $CH_4$ oxidation is on the distribution of $CH_4$ in Indian estuaries.
Additionally, methanogenesis may also be linked with water column stratification (Koné et
al., 2010; Borges and Abril, 2011). In the Indian estuaries, salinity stratification is reported
only during the dry season but it remains active for a small-time span (~2 – 3 weeks) as




evident from daily observations in the Godavari (Sarma et al., 2010), Mandovi and Zuari
(Pedneker et al., 2011), and Krishna estuaries (Dr. T.R. Kumari personal communication,
2016). Thus, methanogenesis in Indian estuaries is likely not principally linked with
stratification (Rao et al., 2016).
Dams and reservoirs are considered a hotspot for methanogenesis. During the initial
phase of reservoir construction (e.g., first decade), $CH_4$ inputs to the river and subsequently to
estuaries can be substantial (Abril, et al., 2005; Kemenes et al., 2007; Kemenes et al., 2011;
Barros et al., 2011). Several dams have been constructed in Indian rivers that store water for
over 6 months (January to June) to meet irrigation, hydropower generation and domestic
needs (Rao et al., 2016). Rao et al. (2016) reported ~3 times higher $CH_4$ levels during the
storage period, indicating significant $CH_4$ production during this time. Monsoonal $CH_4$
concentrations in the Godavari estuary of 72 nM have been reported, which is close to that of
discharge water from the upstream river (73 ± 10 nM; Rao et al., 2016). Additionally, our
analysis shows a positive relationship between $CH_4$ concentration and the number of dams
(Fig. S20), suggesting dams and reservoirs may substantially influence the $CH_4$ budget of
Indian estuaries.
***Air-water $CH_4$ exchange***
During the wet season, the flux of $CH_4$ ($FCH_4$) in the BB estuaries varies between
0.01 and 11.80 µmol m$^{-2}$ d$^{-1}$ (mean: 2.93 ± 4.10 µmol m$^{-2}$ d$^{-1}$; peak in the Ponnayaar). Wet
season $FCH_4$ in the AS estuaries is ~13 times higher (0.11 – 299 µmol m$^{-2}$ d$^{-1}$; peak in the
Tapti estuary) than BB estuaries. However, the dry season shows a contrasting trend with
higher fluxes in the BB estuaries (BB estuaries: 0.08 – 156 µmol m$^{-2}$ d$^{-1}$, peak in the Matla
estuary; AS estuaries: 0.30 – 29.30 µmol m$^{-2}$ d$^{-1}$, peak in the Kochi Backwaters). Positive
fluxes occur throughout both the wet and dry seasons suggesting that the Indian estuaries are
persistent sources of $CH_4$ to the atmosphere.
**Contribution of Indian estuaries to global C budgets**
***Impact on marine C budgets***
A schematic diagram presenting dissolved and particulate C fluxes to/from the Indian
estuaries is presented in the Figure 7. Krishna et al. (2015, 2019) quantified DIC and DOC





export fluxes from Indian estuaries to the northern Indian ocean of ~10.30 Tg C yr$^{-1}$ (~76%
discharges via the BB estuaries) and 2.37Tg yr$^{-1}$ (~30% higher export by BB estuaries than
AS), respectively. Integrating DIC and DOC export fluxes, total dissolved C export via BB
and AS estuaries to the northern Indian ocean is ~12.67 Tg C yr$^{-1}$ of which ~81% is DIC.
From a global perspective, a compilation of current global riverine C export to the ocean is
presented in Table 5. We estimate average global riverine DIC and DOC exports to the ocean
of 393 Tg C yr$^{-1}$ and 218 Tg C yr$^{-1}$, respectively, for a total dissolved C export of 611 Tg C
yr$^{-1}$. Indian rivers constitute ~1.3% of global freshwater discharge; the BB and AS
cumulatively contribute 2.62% and 1.09% to global riverine DIC and DOC export to the
ocean, respectively. This contribution is much lower compared to South American and
African rivers (~17% and ~21% for DIC and DOC, respectively) draining into the Tropical
Atlantic Ocean (Araujo et al., 2014). In total, the BB and AS estuaries contribute 2.07% of
global total dissolved C export to the ocean. The Indian rivers contribute more DIC export
relative to discharge compared to other global rivers (Table 5). The higher DIC flux from the
Indian estuaries links with relatively significant silicate and carbonate mineral weathering
rates in their drainage basins (Gurumurthy et al., 2012; Pattanaik et al., 2013) which in turn is
a function of variability in lithological characteristics and climatology (Huang et al., 2012).

Vegetated coastal wetlands in the Indian subcontinent also play a significant role in

coastal ocean C dynamics. Despite the Indian mangroves covering only ~4% of global
mangrove surface area, C export fluxes from most of the mangroves surrounding estuaries are
not available to precisely understand their impact on the oceanic C budget. In the Indian
Sundarbans, Ray et al. (2018) reported total DIC and DOC export from the Indian
Sundarbans to the BB of ~3.69 and 3.03 Tg C yr$^{-1}$, respectively, for a total of 6.72 Tg C yr$^{-1}$.
The exported DIC from the Indian Sundarban mangroves is ~47% of total DIC export from
the BB, highlighting the large role of the Indian Sundarbans in the DIC budget of BB.
However, the fact that DOC export from the Sundarbans is greater than DOC export from BB
estuaries (~1.34 Tg C yr$^{-1}$) suggests that substantial amounts of DOC are removed in transit
from the Sundarbans estuaries to the continental shelf region and eventually to the northern
Indian Ocean. On the other hand, it is possible that current C export flux estimates have large



inaccuracies. For example, for the Sundarbans mangroves, literature-based discharge data
was used to calculate the export flux rather than real time data. Regarding POC export, there
are not enough observations of POC export fluxes to accurately calculate total export by
Indian estuaries to the ocean.
***Impact on atmospheric C budget***
Despite having higher $pCO_2$ and $CH_4$ concentrations than other world rivers, we
estimate lower air-water fluxes for the Indian rivers (Tables 1 & 4). This suggests that gas
transfer velocities in the region are generally lower (which is a function of the wind speed
over 10 m height of the river, water temperature, and salinity).
Sarma et al. (2012) calculated the total area of the Indian estuaries (consisting of 14
major, 44 medium and 162 minor estuaries) to be ~27000 $km^2$. Integrating the annual mean
$FCO_2$ (22.41 mmol $m^{-2}$ $d^{-1}$) and $FCH_4$ (20.76 µmol $m^{-2}$ $d^{-1}$) based on the data compiled here
to the entire surface area of the Indian estuaries, total $CO_2$ and $CH_4$ emissions from the
estuaries of the Indian sub-continent are estimated to be ~9718 Gg C $yr^{-1}$ and 3.27 Gg C $yr^{-1}$,
respectively. Our recalculated gas flux estimates from the Indian estuaries are ~28% lower
and ~19% higher for $pCO_2$ and $CH_4$, respectively, compared to previous estimates by Sarma
et al. (2012) and Rao et al. (2016). Indian estuaries cover ~2.54% of the global estuarine
surface area and contribute ~0.67% and ~0.12% to global $CO_2$ and $CH_4$ outgassing,
respectively (Table 6). These estimates suggest a limited contribution of Indian estuaries to
global estuarine $CO_2$ and $CH_4$ fluxes. In terms of anthropogenic greenhouse gas emissions,
India emits 2.8 Gt $CO_{2eq}$ annually of which 79%, 14%, and 5% are contributed by $CO_2$, $CH_4$,
and $N_2O$, respectively (Government of India, 2018 Second Biennial Update Report to the
United Nations Framework Convention on Climate Change "India: Third Biennial Update
Report to The United Nations Framework Convention on Climate Change". Archived (PDF)
from the original on 2021-02-27). Thus, emissions of $CO_2$ and $CH_4$ from the Indian estuaries
only represent ~0.44% and ~0.002% of total Indian anthropogenic C emissions. In this
regard, Frankignoulle et al. (1998) estimated $CO_2$ fluxes from European estuaries of 30 – 60
Tg C $yr^{-1}$, which is ~5 to 10% of total European anthropogenic emissions in 1995. This





suggests that despite having ~17% of the global population, the Indian estuaries minutely
contribute to atmospheric C budgets.

Mangroves are both a large sink (i.e., soil C burial) and source of greenhouse gases to

the atmosphere (Chauhan et al., 2008; Krithika et al., 2008; Dutta et al., 2013, 2015, 2017;
Barnes et al., 2011; Biswas et al., 2007; Akhand et al., 2021). Compiling a large mangrove
dataset in the east coast of India, Banerjee et al. (2014) estimated mean $CO_2$ and $CH_4$ fluxes
from the mangrove surrounding water to be 20.18 mol m$^{-2}$ yr$^{-1}$ and 0.027 – 17502 mmol m$^{-2}$
yr$^{-1}$, respectively. The area of water surrounding mangroves is not well defined in India, but
surface waters surrounding mangrove systems generally have lower $CO_2$ emission in contrast
to the higher $CH_4$ emissions compared to fluxes reported here for the Indian estuaries.

To quantitatively understand the potential impact of riverine $CO_2$ and $CH_4$ emissions

on regional climate change scenarios, standard procedure is to report gas emissions in tons of
$CO_2$ equivalents, which is a universal unit of measurement used to indicate the global
warming potential of a greenhouse gases, expressed in terms of a global warming potential of
one unit of $CO_2$ (http://www.defra.gov.uk/enviornment/economy/reporting/). Additionally, it
is important to consider the impact of gas emissions over both 20 year and 100 year time
scales as sources and sinks can vary considerably over decadal timeframes (Kirschke et al.,
2013, Neubauer and Megonigal, 2015), whereas C sequestration estimates may be better
represented over 100 year time frames (Gatland et al., 2014). The global warming potential
for atmospheric $CH_4$ is 56 and 21 times higher compared to the $CO_2$ over 20- and 100-years'
time horizon, or 96 and 45 times higher than $CO_2$ if considering sustained-flux global
warming potential (Neubauer and Megonigal, 2015). Using the former values, the global
warming potential of Indian estuaries via cumulative emissions of $CO_2$ and $CH_4$ is calculated
as ~9.90 x 10$^6$ and 9.79 x 10$^6$ Ton $CO_2$-eq for the 20- and 100-year time horizons,
respectively, of which $CH_4$ contributes only ~1.85% and ~0.70%, respectively.   Our review
has highlighted the major governing factors of estuarine C cycling from the Indian sub-
continent qualitatively. However, more detailed and mechanistic observations of the
processes involved in estuarine C cycling is essential for more precise drafting of Indian
estuarine C budgets, which is intricately linked with the global C cycle in a broader sense.





## Conclusion

In this review paper, data for 20 BB and 12 AS estuaries were compiled and reanalysed based on changes in season and marine end members to explore the mechanisms driving estuarine C biogeochemistry in India. The DIC in Indian estuaries is controlled cumulatively by geochemical (carbonate weathering), climatological (degree of precipitation), and hydrological (mixing) factors. Biogeochemically, carbonate dissolution and organic matter respiration control the DIC levels in the Indian estuaries. DOC behaves mostly non-conservatively and DOC - POC interconversion together with DOC mineralization appear to be major drivers for DOC cycling. POC is composed of freshwater algae, $C_3$ plant material, marine organic matter along with anthropogenic inputs in some eastern Indian estuaries. Respiration and methanogenesis appear to play a pivotal role in controlling POC. The $p\mathrm{CO_2}$ is controlled principally by respiration with freshwater discharge only in the BB, however, POC together with methanotrophy and the abundance of dams control $\mathrm{CH_4}$. $F\mathrm{CO_2}$ estimates showed that AS is a persistent $\mathrm{CO_2}$ source to the atmosphere, however, the BB varies between a source and sink. $F\mathrm{CH_4}$ estimates show that Indian estuaries are a $\mathrm{CH_4}$ source throughout both the AS and BB. From a global perspective, the Indian estuaries contribute 2.62% and 1.09% to global riverine DIC and DOC exports to the ocean, respectively. The total $\mathrm{CO_2}$ and $\mathrm{CH_4}$ flux from the Indian estuaries to the atmosphere are estimated as ~9718 Gg yr$^{-1}$ and 3.27 Gg yr$^{-1}$, which contributes to ~0.67% and ~0.12%, respectively, to the global estuarine emissions estimates. Based on the present review, we suggest that a more through investigation on the mechanisms controlling C cycling (including rate quantification) in Indian estuaries is very essential to fill up the data gap in this area of research and also to more precisely draft the C budget of the estuaries around the Indian subcontinent.

## Data availability

All the data used in the paper is adopted from the literature and the authors are thankfully credited. Moreover, the data has been presented graphically in the Fig. 2-6.



## Author contribution

MKD: Designed the paper, analysed data and wrote first draft of the paper; KS and DP: Designed the paper and reviewed the first draft; NDW, TSB and DP: Edited and reviewed the final version.

## Acknowledgements

M.K.D. is thankful to the NCESS for providing a Research Associate Fellowship. Authors are highly thankful to all potential researchers whose data have been used to draft the manuscript and all of them are thankfully credited in respective places of the manuscript.

## Declaration of Competing Interest

The authors declare that they have no known competing financial interests or personal relationships that could have appeared to influence the work reported in this paper.

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





**Table 1: General characteristics of major Indian estuaries.**

| Marine End member | Name of the estuaries | No of dams | Populations (per km²) | Mean annual discharge (m³ s⁻¹) | Tidal amplitude (m) | Annual mean precipitation (mm) | Catchment area (x 10³ km²) |
|---|---|---|---|---|---|---|---|
| **Bay of Bengal** | Haldi | * | * | 1600 | 7.01 | 1582 | * |
| | Hooghly | * | * | 1751 | * | 1582 | 60 |
| | Saptamukhi | * | * | * | * | * | * |
| | Thakuran | * | * | * | * | * | * |
| | Matla | * | * | * | * | * | * |
| | Subarnarekha | 12 | 338 | 392 | * | 1800 | 29.2 |
| | Baitarani | 8 | 324 | 903 | * | 1450 | 14.2 |
| | Rushikulya | 13 | 360 | 61 | * | 1000 | 9 |
| | Mahanadi | 280 | 282 | 2121 | 2.82 | 1406 | 141.6 |
| | Dhamra | * | * | * | * | * | * |
| | Vamsadhara | 3 | 130 | 113 | * | 1400 | 11 |
| | Nagavali | 4 | 150 | 64 | 2.17 | 1000 | 9.4 |
| | Godavari | 978 | 193 | 3505 | 2.1 | 1300 | 313 |
| | Krishna | 736 | 260 | 2213 | 1.98 | 784 | 259 |
| | Penna | 61 | 186 | 200 | * | 510 | 186 |
| | Veller | 3 | 457 | 29 | 1.51 | 980 | 457 |
| | Ponnayaar | 4 | 291 | 51 | * | 969 | 291 |
| | Cauvery | 122 | 393 | 677 | * | 1075 | 393 |
| | Ambalayaar | * | * | 28 | * | * | * |
| | Vaigai | 2 | 499 | 36 | * | 850 | 7 |
| **Arabian Sea** | Kochi Back water | * | * | 391 | 1.34 | * | * |
| | Chalakudi | 6 | * | 60.88 | * | 3600 | 1.7 |
| | Bharatakulza | 13 | * | 161 | * | 2500 | 6.2 |
| | Netravathi | * | 103 | 351 | * | 3923 | 3.2 |
| | Sharavathi | 3 | 109 | 144 | * | 4000 | 3.6 |
| | Kali | 6 | 111 | 152 | * | 3200 | 4.2 |
| | Zuari | 3 | 92 | 103 | 2.7 | 3500 | 1 |
| | Mandovi | 2 | 62 | 105 | 2.7 | 3500 | 3.6 |
| | Narmada | 281 | 184 | 1447 | 10.9 | 1120 | 99 |
| | Tapti | 375 | 208 | 472 | * | 888 | 65 |
| | Sabarmathi | 62 | 1702 | 120 | * | 787 | 21.7 |
| | Mahisagar | 138 | 507 | 349 | 7.63 | 785 | 34.8 |










**Table – 2: The DIC contents in some of the major estuaries of the world.**

| Rivers | DIC (µM) | $p$CO$_2$ (µatm) | $F$CO$_2$ (mmol m$^{-2}$ d$^{-1}$) | References |
|---|---|---|---|---|
| **Mississippi** | 540 | 1335 | 270 | Li et al., 2013 |
| **Amazon** | ** | 4350 | 189 | Richey et al., 2002 |
| **Hudson** | ** | 1125 | 1637 | Li et al., 2013 |
| **Yangtze** | 1700 | 1297 | 14.2 | Wang et al., 2007 |
| **St. Lawrence** | 460 | 1300 | 78-295 | Li et al., 2013 |
| **Xi river** | 1580 | 2600 | 190-357 | Yao et al., 2007 |
| **Ottawa** | 50-300 | 1200 | 81 | Telmar et al., 1999 |
| **Mekong** | 1590 | 1090 | 195 | Li et al., 2013 |
| **Maotiao** | 2600 – 3020 | 3740 | 295 | Wang et al., 2011 |
| **Pearl River** | 1850 - 3329 | 168-8364 | -25.82 – 2293.58 | Guo et al., 2008, 2009 |
| **Artic rivers** | 642-1792 | ** | ** | Tank et al., 2012 |
| **Tyne** | 1208-3867 | ** | ** | Baker and Inverarity (2004) |
| **Ouseburn** | 2592 - 5317 | ** | ** | Baker (2002) |
| **River Tern** | 1742-3242 | ** | ** | Cumberland and Baker (2007) |
| **Columbia** | ** | 176-735 | -53 – 193.2 | Evans et al. 2013 |
| **Indian estuaries** | **280-4166** | **248-15220** | **-4.67 – 96.32** | **This review work** |

**Data not available











**Table – 3: The DOC contents in some of the major estuaries of the world.**

| Rivers | DOC (µM) | References |
|---|---|---|
| Amazon Mainstream | 300 | Richey et al., 1990 |
| St. Lawrence | 313 | Pocklington and Tan, 1987 |
| Elbe | 325-500 | Ludwig et al., 1997 |
| Nile | 246 | Abu el Ella., 1993 |
| Columbia | 177 | Damn et al., 1981 |
| Yellow river | 267-708 | Zhang et al., 1992 |
| Rone | 144 | Kempe et al., 1991 |
| Delware and Hudson | 12.9 - 46.4 | Seitzinger and Sanders, 1997 |
| Mississippi | 489 | Bianchi et al., 2001 |
| Ganga-Brahmaputra | 323 | Safiullah et al., 1987 |
| Congo | 604 | Probst and Suchet, 1992 |
| Yangtze | 167 - 842 | Zhang et al., 2005 |
| Rioni | 88 | Romankevich and Artemyev, 1985 |
| Seven | 258 - 650 | Mantoura and Woodward, 1983 |
| Niger | 309 | Martins and Probst, 1991 |
| Artic rivers | 7.9 - 65 | Holmes et al., 2012; Letscher et al., 2013 |
| **Indian estuaries** | **37 - 1079** | **This review work** |

*Mentioned earlier; **Data not available

**Table – 4: The CH$_4$ fluxes in some of the major estuaries of the world.**

| Rivers | CH$_4$ (nM) | FCH$_4$ (µmol m$^{-2}$ d$^{-1}$) | References |
|---|---|---|---|
| Pearl River estuary | 7-174 | 63.5 | Zhou et al., 2009 |
| Tyne | 13-654 | ** | Upstill-Goddard et al., 2000 |
| European estuaries | 2-3600 | 130 | Middelburg et al., 2002 |
| Humber | 16-669 | ** | Upstill-Goddard et al., 2000 |
| Hudson | 50-940 | 350 | De Angelies and Scranton, 1993 |
| Brisbane | 31 - 578 | 19-1725 | Musenze et al., 2014 |
| Danube | 131 | 470 | Amouroux et al., 2002 |
| Yangtze | 13-27 | 35-144 | Zhang et al., 2008 |
| Bodden | 2.4-370 | 30-210 | Bange et al., 1998 |
| Ivory Coast | ** | 25-1187 | Kone et al., 2010 |
| Choptank river estuary | ** | 2400 | Lipschultz (1981) |
| Rhine and Scheldt | 2.5-370 | 6-600 | Scranton and McShane (1991) |
| Tomales Bay | 8-100 | 7- 10 | Sansone et al. (1998) |
| Temmesjoki estuary | 240-506 | ** | Silvennoinen et al. (2008) |
| Randers Fjord estuary | 41-420 | 70-410 | Abril and Iversen (2002) |
| Rio San Pedro, | 12-87 | 34-150 | Ferrón et al. (2007) |
| Artic rivers | ** | 80-1020 | Kling et al., 1992 |
| **Indian estuaries** | **4 - 573** | **0.01-299** | **This review work** |

**Data not available


**Table – 5: Contribution of Indian estuaries to global estuarine C export to the ocean. The table is modified after Li et al. (2017). Values given here are in Tg yr⁻¹ unit. *Data not available; **Calculation not possible; TDC = Total dissolved C, TPC = Total particulate C, TC = Total C, $C_{DIC}$ = Contribution of DIC in TDC.**

| Export flux | Meybeck (1982, 1987) | Luwding et al. (1996a, b) | Harrison et al. (2005); Beusen et al. (2005) | Cai (2011) | Li et al. (2017) | Mean global riverine C export | Export from (BBE + ASE) | Global contributions by (BBE + ASE) |
|---|---|---|---|---|---|---|---|---|
| **DIC** Export | 430 | 320 | * | 410 | 410 | 393 | 10.30 | 2.62% |
| **DOC** Export | 220 | 210 | 170 | 250 | 240 | 218 | 2.37 | 1.09% |
| **TDC** Export = **(DIC + DOC)** Export | **650** ($C_{DIC}$ ~66%) | **530** ($C_{DIC}$ ~60%) | ** | **670** ($C_{DIC}$ ~61%) | **650** ($C_{DIC}$ ~63%) | **611** ($C_{DIC}$ ~64%) | **12.67** ($C_{DIC}$ ~81%) | **2.07%** |
| **POC** Export | 180 | 170-190 | 200 | 220 | 240 | 204 | * | ** |
| **PIC** Export | 170 | 170 | * | 170 | 170 | 170 | * | ** |
| **TPC** Export = **(PIC + POC)** Export | 350 | ~350 | ** | 390 | 410 | 374 | ** | ** |
| **Total C export =** **(TDC + TPC)** Export | 1000 | 880 | ** | 1050 | 1060 | 985 | ** | ** |

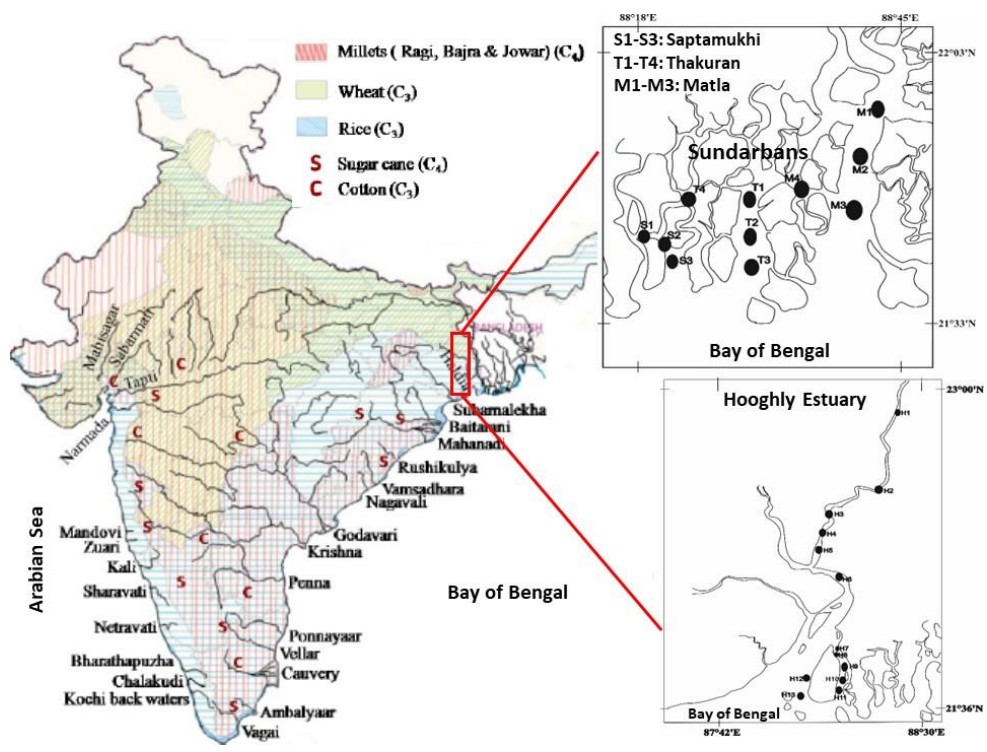


Fig. 1: Locations of the major estuaries of India. Modified from Sarma et al. (2012) and Dutta
et al. (2019a).




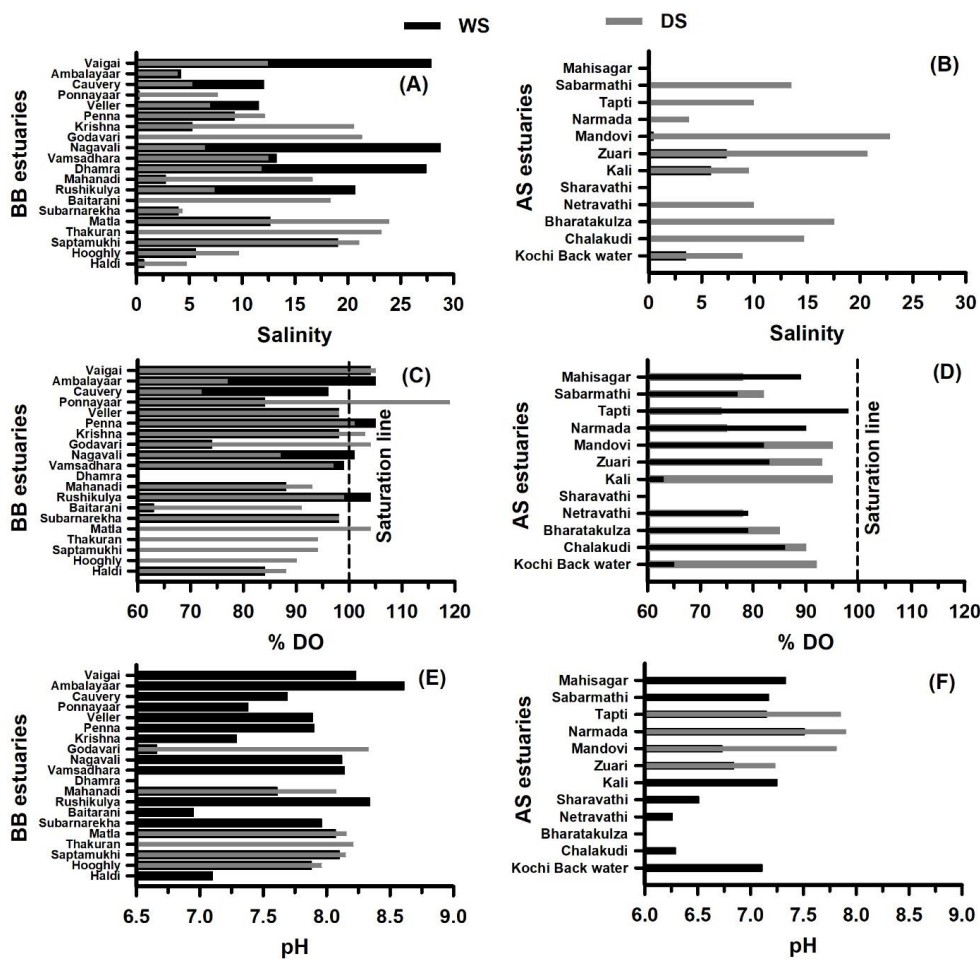


Fig. 2: (a) Salinity, (b) %DO, and (c) pH for the major Indian estuaries. WS = wet season; DS
= dry season





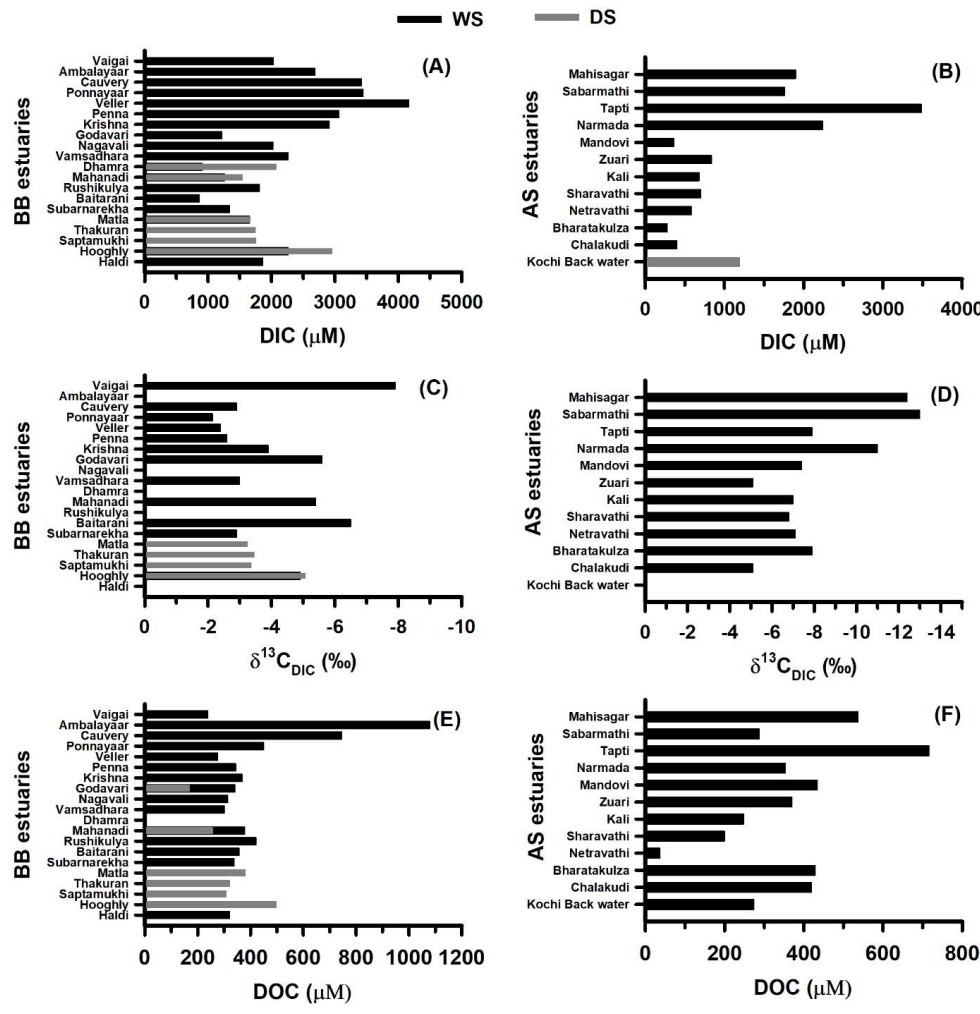

Fig.3: Variabilities of DIC, δ¹³C_DIC and DOC in the major Indian estuaries. For the Hooghly, Saptamukhi, Thakuran and Matla, mean of data reported by Dutta et al. (2019a, 2021) during pre- and post-monsoon are used as dry season data. In other cases, mean data is used where multiple data is available. WS = wet season; DS = dry season.




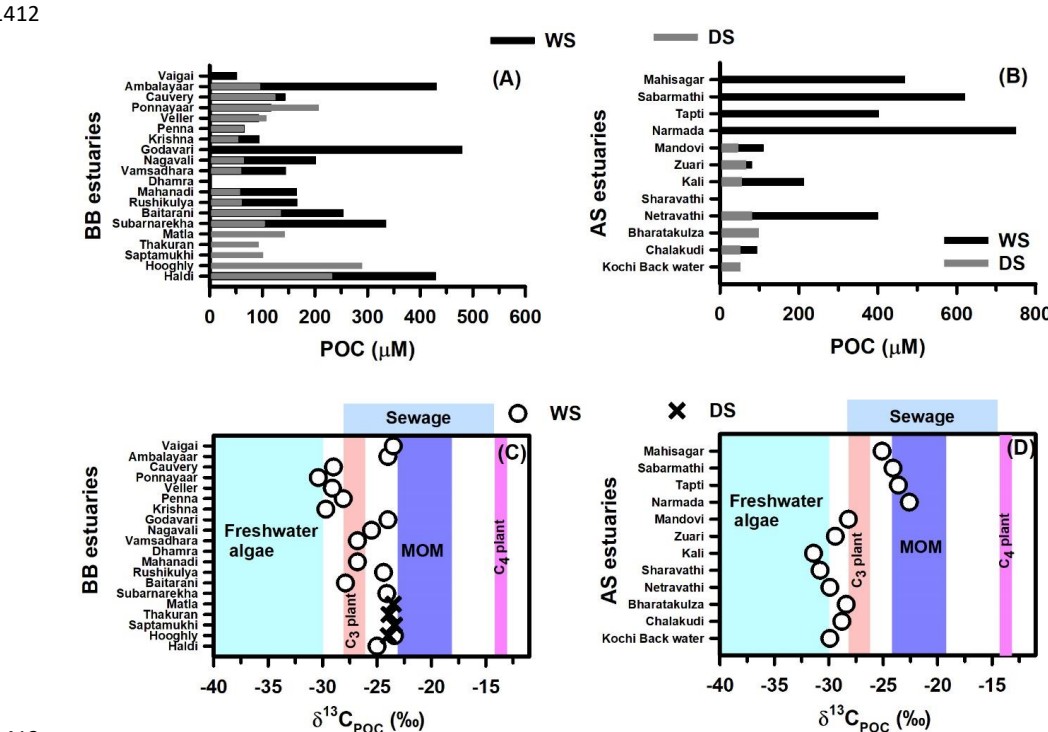


Fig.4: (A) POC distribution in the BB estuaries, (B) POC distribution in the AS estuaries, (C)
$\delta^{13}C_{POC}$ in the BB estuaries, and (D) $\delta^{13}C_{POC}$ in the AS estuaries. WS = wet season; DS = dry
season; MOM = marine-derived organic matter.


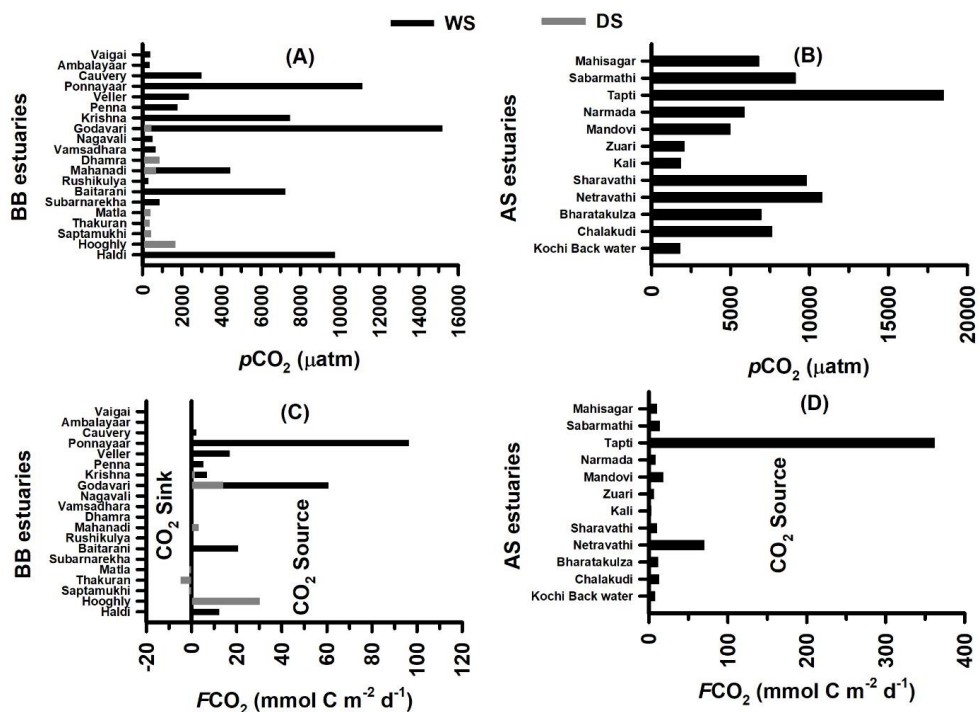

Fig.5: (A) $pCO_2$ in the BB estuaries, (B) $pCO_2$ in the AS estuaries, (C) $FCO_2$ in the BB estuaries, and (D) $FCO_2$ in the AS estuaries. WS = wet season; DS = dry season



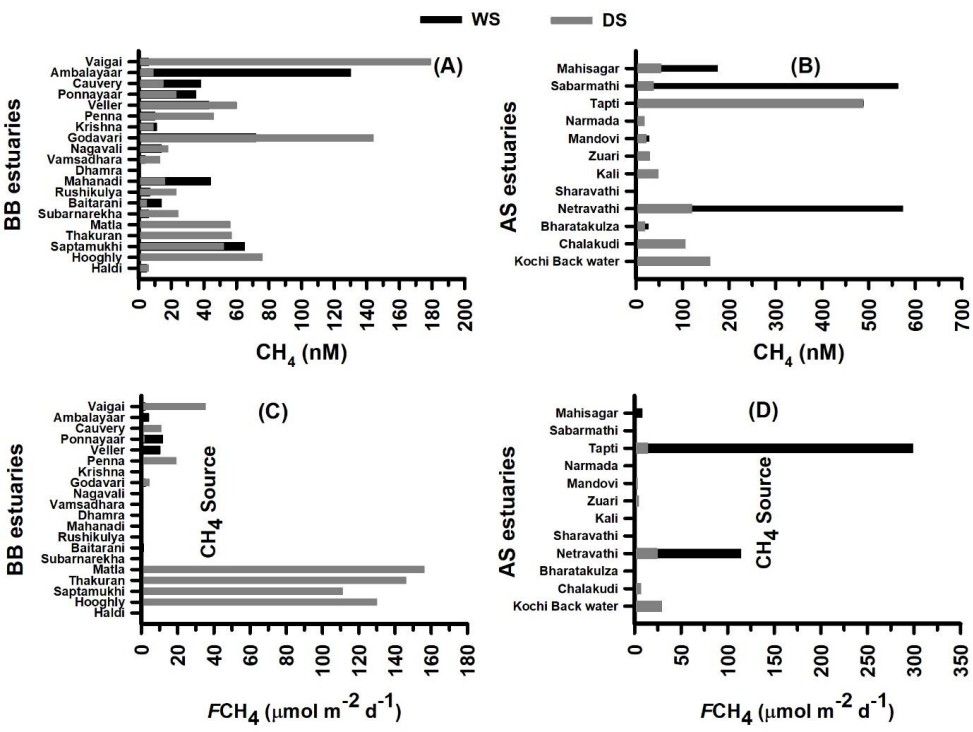


Fig. 6: (A) $CH_4$ in the BB estuaries, (B) $CH_4$ in the AS estuaries, (C) $FCH_4$ in the BB
estuaries, and (D) $FCH_4$ in the AS estuaries. WS = wet season; DS = dry season















**Atmosphere**

$9718$ **Gg** $CO_2$ **yr**$^{-1}$   $3.27$ **Gg** $CH_4$ **yr**$^{-1}$

Fig. 7: A schematic diagram presenting dissolved and particulate C fluxes to/from the
estuary. Magnitude is presented where available.









