# Peer review of "Reviews and Syntheses: Carbon biogeochemistry of Indian estuaries"

_Biogeosciences, 2022_

## Referee Comment (RC1)

**Overview**

The authors reviewed carbon biogeochemistry in the India Coast, presented its spatiotemporal variability and discussed the potential drivers. As a continuum between land and ocean, estuarine system is important while suffering a great spatiotemporal heterogeneity. Such study could help constrain this variation and better understand its essential role on global carbon budget. However, this manuscript of its current version still saves a large space for improvement, particularly from its data interpretation and content structure perspectives. I am afraid this manuscript will need a thorough revision to fit for the journal.

**Major comments**

(1) Traceable data is vital for a review article. Unfortunately, there is no clear pathway(s) for data sources in this manuscript. For example, how many sampling stations, how many observations for each estuary, sampling time, etc.? We even do not know if the authors are presenting annual average data or just one-time surveyed data. Why there is no standard deviation for each estuary data in figures? A proper conclusive data table (can be supplementary material) is badly needed to show its rigor and reliability.

(2) Following the first major concern, then the data interpretation is problematic. First, the data visualization needs improvement, why only list Sundarbans and Hooghly Estuary sampling stations in Fig.1? Differences on estuaries or dry/wet cycle cannot be well distinguished in both figures and supplementary figures. Second, the way of data processing is also unclear, for example, how do the authors conduct statistical analysis, t-test? two-way ANOVA? any process to meet the assumptions? In supplementary figures, several estuaries are excluded to meet a high p-value relationship seems arbitrary and misleading, same as the threshold 6800 μatm for $p\text{CO}_2$. Is there any reason/accordance to do so?

Also, I have a feeling that the authors messed with riverine and estuarine data. For example, in Line 815 the "~10.30 Tg C yr$^{-1}$" belongs to riverine export fluxes (Krishna et al., 2019) rather than "export fluxes from Indian estuaries", and the following discussion (Lines 819–831) is all about riverine C exports. Accordingly, in Fig. 7 export flux values may put in wrong place. Similarly, I do not think there are so many dams built in coastal estuaries list in Table 1. This is the reason why readers are curious about the data details, if so, I would suggest the authors clarify each estuary area/coordinates and further check about the data.

(3) The manuscript structure is organized in a research article format instead of a review. In addition, the separated discussions on DIC, DOC, POC, $CO_2$, $CH_4$ read super repetitive and distracting. In fact, carbon biogeochemistry is comprehensive and synthesized, drivers (e.g. hydrologic, biochemical, etc.) on any single carbon species would further impact on other carbon interactivities and then the entire carbon budget. Re-organization of manuscript structure to look at the drivers more synthetically is highly recommended.

(4) Many important information are missing, such as temperature gradient, wind speeds, net

ecosystem productions, submarine groundwater discharge rates, two end-members values, etc., these are decisive to estuarine carbon biogeochemistry. Also, I am curious about the anthropogenic impact on estuarine carbon biogeochemistry. It seems the anthropogenic discharges in this study are mostly referred as sewage discharges to upper rivers, then how to identify the anthropogenic carbon in lower estuarine area proportionally?

**Line comments**

Line 210: more details on "statistical analysis".

Line 347: references.

Line 348: "DIC addition/removal" details.

Line 399: should be "riverine DIC" instead of "estuarine DIC"

Line 483: the difference between "Terrestrial DOC" and "Riverine DOC" ?

Line 492: where is "Fig. 12A"?

Line 540-543: the purpose for comparing regional DOC/DON to POC/PON? or DOC fraction in global coastal ocean?

Line 552-553: you cannot say this unless the data about POC/DOC from two end-members.

Line 570: why 6800 μatm threshold?

Line 579: further explain "a decrease of aerobic bacterial activity with increasing DOC"

Line 616: further explain "freshwater mixing is not the major driver of POC", as it shows lower salinity with higher POC and 13C values.

Line 668-671: more direct evidence is needed to evaluate anthropogenic impact on $pCO_2$ rather than population density. For example, anthropogenic $pCO_2$ is 100 μatm out of total $pCO_2$ 400 μatm in Estuary A, where as anthropogenic $pCO_2$ is 200 μatm out of total $pCO_2$ 1000 μatm in Estuary B.

Line 686: where is "Fig. 21"?

Line 695-696: wrong statement "nitrification plays crucial role in increasing pH"

Line 699: "unlikely"

Line 712-713: details for FCO2

Link 862: where is "Table 6"

**Tables and Figures**

Table 1: add coordinates, references

Table 2: are they annual averaged numbers? Standard deviation?

Table 5: confusing table, please improve

Fig. 1: why only zoom in two estuaries? Instead display C3 and C4 plants area, population density is more important to be visualized.

Fig. 2 - Fig. 6: cannot distinguish that data between dry and wet, standard deviation needed.

Fig. 6: estuarine export fluxes values should be river-borne C, the figure is unnecessary if only two

components are evaluated.

For all supplementary figures: there is no spatial information, reason why exclude several estuarine data, the number of observations are too small, standard deviations? Data interpretation seems unconvincing due to potential data manipulation.

---

## Author Comment (AC1)

**Response to reviewer – 1**

**Overview**

The authors reviewed carbon biogeochemistry in the India Coast, presented its spatiotemporal variability and discussed the potential drivers. As a continuum between land and ocean, estuarine system is important while suffering a great spatiotemporal heterogeneity. Such study could help constrain this variation and better understand its essential role on global carbon budget. However, this manuscript of its current version still saves a large space for improvement, particularly from its data interpretation and content structure perspectives. I am afraid this manuscript will need a thorough revision to fit for the journal.

*Response: Thanks for the encouraging comments about the manuscript and valuable suggestions. We have included your suggestions in the revised manuscripts to improve the quality of the manuscript.*

**Major comments**

(1) Traceable data is vital for a review article. Unfortunately, there is no clear pathway(s) for data sources in this manuscript. For example, how many sampling stations, how many observations for each estuary, sampling time, etc.? We even do not know if the authors are presenting annual average data or just one-time surveyed data. Why there is no standard deviation for each estuary data in figures? A proper conclusive data table (can be supplementary material) is badly needed to show its rigor and reliability.

*Response: We agree and have added an excel file comprising all data used to prepare the manuscript along with the references as supplementary file. In the file, we have compiled all information related to that data wherever available.*

(2) Following the first major concern, then the data interpretation is problematic. First, the data visualization needs improvement, why only list Sundarbans and Hooghly Estuary sampling stations in Fig.1? Differences on estuaries or dry/wet cycle cannot be well distinguished in both figures and supplementary figures. Second, the way of data processing is also unclear, for example, how do the authors conduct statistical analysis, t-test? two-way ANOVA? any process to meet the assumptions? In supplementary figures, several estuaries are excluded to meet a high p-value relationship seems arbitrary and misleading, same as the threshold 6800 μatm for $pCO_2$. Is there any reason/accordance to do so?

*Response: The Hooghly and Sundarbans are separately shown in Fig. 1. As you can see three major estuaries (Saptamukhi, Thakuran, and Matla) are included within the Sundarbans. Additionally, the Hooghly and estuaries of Sundarbans are closely associated which are covered thoroughly by Dutta et al., (2019, 2021) from the upstream to downstream and mean data is used in the paper. Considering limited space in the given map and inclusion of the sampling points, the Hooghly-Sundarbans system was plotted as a sub-set in Fig. 1.*

*Regarding figures, we have now revised the figures for the revised manuscript. In the original manuscript, we used simple regression analysis to evaluate dependency between C and other parameters. But, yes, as you said each parameter may be related, in the revised manuscript we have included 't' test and PCA analysis wherever applicable as follows:*

*Table – Results for statistical 't' test analysis between the mean of available data. The analysis is performed at 95% confidence level.*

| Parameters | Inter-seasonal comparison | | Inter-BB & AS estuaries comparison | |
|---|---|---|---|---|
| | **BB estuaries** | **AS estuaries** | **Wet Season** | **Dry Season** |
| Salinity | $p = 0.55$ | $p < 0.0001$** | $p = 0.003$** | $p = 0.84$ |
| %DO | $p = 0.60$ | $p = 0.31$ | $p = 0.07$ | $p = 0.001$** |
| pH | $p = 0.10$ | $p = 0.006$** | $p < 0.0001$** | $p = 0.012$** |
| DIC | $p = 0.59$ | NA | $p = 0.008$** | NA |
| $\delta^{13}C_{DIC}$ | $p = 0.69$ | NA | $p = 0.000$** | NA |
| DOC | $p = 0.32$ | NA | $p = 0.45$ | NA |
| POC | $p = 0.02$** | $p = 0.02$** | $p = 0.17$ | $p = 0.06$ |
| $\delta^{13}C_{POC}$ | $p = 0.04$** | NA | $p = 0.21$ | NA |
| $pCO_2$ | $p = 0.07$ | NA | $p = 0.10$ | NA |
| $FCO_2$ | $p = 0.41$ | NA | $p = 0.29$ | NA |
| $CH_4$ | $p = 0.39$ | $p = 0.38$ | $p = 0.03$** | $p = 0.11$ |
| $FCH_4$ | $p = 0.05$** | $p = 0.27$ | $p = 0.13$ | $p = 0.16$ |

*\*\*Statistically significant at $\alpha = 0.05$.*

*Principal component analysis (PCA) was performed in order to identify major controlling factors for dissolved and particulate C as well as variability of trace gases ($CO_2$ and $CH_4$) in the major estuaries of India. The PCA was performed for 7 parameters (DIC and its isotopic composition, DOC, POC and its isotopic compositions, $pCO_2$ and $CH_4$ concentration) based on the availability of other parameters (no of dams, population density, precipitation, estuarine area, discharge, catchment area, salinity, %DO, and pH). Moreover, due to scarcity of data during the dry season the PCA analysis is restricted only during the wet season. The principal component (PC) with eigen values >1 was considered for further analysis and only two factors were identified in this case as given below:*

*Table: Results for the Principal Component Analysis of estuarine carbon biogeochemistry of India.*

| | BB estuaries | | AS estuaries | |
|---|---|---|---|---|
| Eigen value | 2.90 | 2.07 | 4.40 | 1.65 |
| Explained variance (%) | 41.48 | 29.65 | 62.83 | 23.58 |
| Cumulative | 41.48 | 71.11 | 62.63 | 86.41 |
| **Variable** | **PC1** | **PC2** | **PC1** | **PC2** |
| DIC | 0.89 | 0.14 | 0.92 | 0.21 |
| $\delta^{13}C_{DIC}$ | 0.73 | 0.27 | -0.65 | 0.67 |
| DOC | 0.40 | 0.55 | 0.58 | 0.72 |
| POC | -0.76 | 0.46 | 0.76 | -0.61 |
| $\delta^{13}C_{POC}$ | -0.82 | -0.35 | 0.92 | -0.21 |
| $pCO_2$ | -0.35 | 0.80 | 0.87 | 0.46 |

| | | | | |
|---|---|---|---|---|
| *CH₄* | *-0.23* | *0.84* | *0.78* | *0.12* |

*The PC1 accounts for ~41.48% and 62.63% variability for the BB and AS estuaries with strong factor loadings of DIC, $\delta^{13}C_{DIC}$, POC and $\delta^{13}C_{POC}$ for BB estuaries and DIC, $\delta^{13}C_{DIC}$, DOC, POC, $\delta^{13}C_{POC}$, $pCO_2$ and $CH_4$ for the AS estuaries. For the BB estuaries, the factor loading might be links with the biological productivity; however, for the AS estuaries the factor loading might be due to POC and DOC decomposition and its associated productions of $CO_2$ and $CH_4$. The oxygenated estuary supports aerobic degradation of organic matter producing $CO_2$ and considered to be restrain anaerobic degradation. However, it should also be noted that $CH_4$ production in the isolated anoxic microhabitats of sinking particulate organic matter, in well-oxygenated water column, have been observed in the open ocean (see Reeburgh, 2007). The PC2 represents 71.11% and 86.41% of total variance for the BB and AS estuaries with relatively strong factor loading for DOC, $pCO_2$ and $CH_4$ for the BB estuaries and $\delta^{13}C_{DIC}$, DOC, and POC for the AS estuaries. The former might be associated with aerobic and anaerobic degradations of DOC, however, the later might be linked with aerobic decomposition of DOC and POC and their impact on variability of $\delta^{13}C_{DIC}$.*

*Thereafter, Pearson correlation analysis was performed for the factor loadings of the estuarine data with the number of dams, estuarine area, population density, mean annual discharge, catchment area, salinity, pH and %DO. The Pearson correlation matrix for the BB estuaries suggests PC1 is strongly controlled by area of the estuary while PC2 is controlled by the cumulative interactions between salinity, pH as well as catchment area. For the AS estuaries, despite the PC1 is controlled by no of dams, catchment area, salinity and %DO but the PCA analysis failed to highlight any controlling factor for the PC2.*

*Correlation Matrix (Pearson) for the BB estuaries:*

| Variables | Dams | Population (/km2) | Area | Discharge | Precipitation | Catchment area (x 10³ km2) | Salinity | %DO | pH | PC1 | PC2 |
|---|---|---|---|---|---|---|---|---|---|---|---|
| Dams | **1** | -0.373 | **0.805** | **0.934** | -0.052 | 0.318 | -0.368 | -0.272 | **-0.675** | -0.420 | 0.528 |
| Population(/km2) | -0.373 | **1** | -0.373 | -0.362 | -0.083 | 0.155 | 0.490 | 0.161 | 0.293 | 0.108 | -0.209 |
| Area | **0.805** | -0.373 | **1** | **0.766** | 0.128 | 0.282 | -0.370 | -0.439 | **-0.675** | **-0.605** | 0.596 |
| Discharge | **0.934** | -0.362 | **0.766** | **1** | 0.169 | 0.194 | -0.496 | -0.464 | **-0.743** | -0.570 | 0.541 |
| Precipitation | -0.052 | -0.083 | 0.128 | 0.169 | **1** | -0.402 | -0.376 | -0.429 | -0.135 | -0.547 | 0.047 |
| Catchment area (x 10³ km2) | 0.318 | 0.155 | 0.282 | 0.194 | -0.402 | **1** | -0.181 | 0.030 | -0.273 | 0.465 | **0.720** |
| Salinity | -0.368 | 0.490 | -0.370 | -0.496 | -0.376 | -0.181 | **1** | **0.680** | **0.748** | 0.175 | **-0.657** |
| %DO | -0.272 | 0.161 | -0.439 | -0.464 | -0.429 | 0.030 | **0.680** | **1** | **0.848** | 0.516 | -0.539 |
| pH | **-0.675** | 0.293 | **-0.675** | **-0.743** | -0.135 | -0.273 | **0.748** | **0.848** | **1** | 0.412 | **-0.741** |
| PC1 | -0.420 | 0.108 | **-0.605** | -0.570 | -0.547 | 0.465 | 0.175 | 0.516 | 0.412 | **1** | 0.000 |
| PC2 | 0.528 | -0.209 | 0.596 | 0.541 | 0.047 | **0.720** | **-0.657** | -0.539 | **-0.741** | 0.000 | **1** |

*Values in bolded digits are statistically significant at alpha = 0.05*

*Correlation Matrix (Pearson) for the AS estuaries:*

| Variables | Dams | Population (/km2) | Area | Discharge | Catchment area (x 103 km2) | Salinity | %DO | pH | PC1 | PC2 |
|---|---|---|---|---|---|---|---|---|---|---|
| Dams | **1** | -0.092 | 0.518 | 0.686 | -0.713 | -0.554 | **-0.890** | 0.550 | **0.821** | 0.224 |
| Population (/km2) | -0.092 | **1** | 0.253 | -0.202 | -0.550 | -0.376 | -0.271 | 0.184 | 0.457 | -0.509 |
| Area | 0.518 | 0.253 | **1** | **0.828** | -0.502 | -0.491 | -0.569 | 0.560 | 0.490 | -0.552 |
| Discharge | 0.686 | -0.202 | **0.828** | **1** | -0.455 | -0.380 | -0.645 | 0.696 | 0.412 | -0.353 |
| Catchment area (x 103 km2) | -0.713 | -0.550 | -0.502 | -0.455 | **1** | 0.729 | **0.938** | -0.699 | **-0.940** | 0.265 |
| Salinity | -0.554 | -0.376 | -0.491 | -0.380 | 0.729 | **1** | 0.687 | -0.319 | **-0.777** | 0.177 |
| %DO | **-0.890** | -0.271 | -0.569 | -0.645 | **0.938** | 0.687 | **1** | -0.685 | **-0.935** | 0.100 |

| | | | | | | | | | | |
|---|---|---|---|---|---|---|---|---|---|---|
| pH | 0.550 | 0.184 | 0.560 | 0.696 | -0.699 | -0.319 | -0.685 | **1** | 0.501 | -0.531 |
| PC1 | **0.821** | 0.457 | 0.490 | 0.412 | **-0.940** | **-0.777** | **-0.935** | 0.501 | **1** | 0.000 |
| PC2 | 0.224 | -0.509 | -0.552 | -0.353 | 0.265 | 0.177 | 0.100 | -0.531 | 0.000 | **1** |

*Values in bolded digits are statistically significant at alpha = 0.05*

*In the revised manuscript, without eliminating any data we have included the PCA analysis (as mentioned above) to highlight major controlling factors for estuarine C biogeochemistry. We hope it will be well accepted to the reviewers as well as editors.*

Also, I have a feeling that the authors messed with riverine and estuarine data. For example, in Line 815 the "~10.30Tg C yr$^{-1}$" belongs to riverine export fluxes (Krishna et al., 2019) rather than "export fluxes from Indian estuaries", and the following discussion (Lines 819–831) is all about riverine C exports. Accordingly, in Fig. 7 export flux values may put in wrong place. Similarly, I do not think there are so many dams built in coastal estuaries list in Table 1. This is the reason why readers are curious about the data details, if so, I would suggest the authors clarify each estuary area/coordinates and further check about the data.

*Response: Yes, we have crosschecked the export flux data with the Krishna et al. (2019). The data in Line no – 815 of the manuscript belongs to riverine export fluxes to the northern Bay of Bengal. Thanks for pointing it out. We have removed the value in the revised manuscript.*

*Yes, in Fig. 7 export fluxes have been placed in the wrong location We have decided to remove the figure from the revised manuscript.*

*The salinity of the observed estuaries never reached zero (salinity = 0.04 – 23.91). The paper from where most of the data have been picked up is mentioned the data as for "Indian Monsoonal Rivers" but based on the observed salinity we belief the data points are better represented as estuarine data points (better to say freshwater to mixing regime data points) rather than ideal riverine data and that's why we selected the title of the paper as "Carbon biogeochemistry of the Indian estuaries". But for the Hooghly estuary and estuaries of Sundarbans, data for the freshwater to marine regime data were included.*

*The exact sampling coordinate for all the estuaries are not available from the supportive literature. But based on the availability, we have updated the revised manuscript.*

(3) The manuscript structure is organized in a research article format instead of a review. In addition, the separated discussions on DIC, DOC, POC, $CO_2$, $CH_4$ read super repetitive and distracting. In fact, carbon biogeochemistry is comprehensive and synthesized, drivers (e.g. hydrologic, biochemical, etc.) on any single carbon species would further impact on other carbon interactivities and then the entire carbon budget. Re-organization of manuscript structure to look at the drivers more synthetically is highly recommended.

*Response: Yes, we have noted the points and have improved the revised manuscript accordingly. All C parameters are interlinked; keeping that in mind we have included PCA in the revised manuscript (as described above).*

(4) Many important information is missing, such as temperature gradient, wind speeds, net ecosystem productions, submarine groundwater discharge rates, two end-members values, etc., these are decisive to estuarine carbon biogeochemistry. Also, I am curious about the anthropogenic impact on estuarine carbon biogeochemistry. It seems the anthropogenic

discharges in this study are mostly referred as sewage discharges to upper rivers, then how to identify the anthropogenic carbon in lower estuarine area proportionally?

*Response: Agreed. All the information is inherently linked with estuarine carbon biogeochemistry. Unfortunately, all the above-mentioned chapters/ parameters have not been thoroughly examined in the estuarine carbon biogeochemistry research in India. Nevertheless, we have tried to better incorporate the above-mentioned factors as much as possible in the revised manuscript.*

*Regarding two-end members mixing model analysis, despite Bouillon et al. (2003) and thereafter Samanta et al. (2015) and Dutta et al. (2019, 2021) identified some major governing factors for estuarine carbon biogeochemistry, however, the two end members seasonal values for all the Indian estuaries are not available which precluded us form applying these more broadly across all the data.*

*Regarding the degree of anthropogenic impact, we simply do not have enough information to comprehensively discuss its impact. This is why we used population density as a proxy for anthropogenic inputs in the paper. For this reason, we have de-emphasized the discussion of anthropogenic inputs by changing this section into a paragraph combined with a new "natural and anthropogenic sources" section.*

*Yes, the reported $\delta^{13}C_{POC}$ values provided a signal for sewage inputs in the Indian estuaries and the sewage is mainly discharged in the freshwater region of the estuary (like the Hooghly). However, it may be quite difficult to identify it in the lower estuarine area as the possibility exists for its biogeochemical modification within the upper to the lower estuarine stretch. Due to that modification the anthropogenic signal might be masked in the lower estuary and the same has been reported in the anthropogenically stressed Hooghly estuary (Dutta et al. 2019, 2021).*

**Line comments**

Line 210: more details on "statistical analysis".

*Response: As advised 't' test and PCA test are included in the revised manuscript. It is briefly discussed in the methods section of the revised manuscript.*

Line 347: references.

*Response: Added references.*

Line 348: "DIC addition/removal" details.

*Response: Depending upon the physicochemical and micromaterial condition of the estuary, the DIC addition includes organic matter respiration and carbonate dissolution whereas DIC removal includes $CO_2$ outgassing, phytoplankton productivity and carbonate precipitation. We have included it in the revised manuscript.*

Line 399: should be "riverine DIC" instead of "estuarine DIC".

*Response: Yes, "riverine DIC" will be a much better term to use here. We have replaced it in the revised manuscripts.*

Line 483: the difference between "Terrestrial DOC" and "Riverine DOC"?

*Response: In aqueous systems, DOC can be added both internally as well as externally. The DOC originating from within the river is known as autochthonous DOC and typically comes from aquatic plants or algae. However, DOC originating outside the river is known as allochthonous DOC which typically comes from soils or terrestrial plants and ultimately discharges to the river.*

Line 492: where is "Fig. 12A"?

*Response: Sorry, it is s typographical error. We have removed it from the revised manuscript. Thanks for pointing out the mistake.*

Line 540-543: the purpose for comparing regional DOC/DON to POC/PON? or DOC fraction in global coastal ocean?

*Response: It was included just to compare fractions of C and N in the dissolved and particulate forms. However, we have removed it from the revised manuscript.*

Line 552-553: you cannot say this unless the data about POC/DOC from two end-members.

*Response: Yes, agree. In the revised manuscript "line no 548-558 of the pre-revised manuscript" is modified as "The DOC was dominant over POC throughout all Indian estuaries (DOC/POC >1). Additionally, a recent study on the eastern Indian estuaries showed DOC and POC inter-conversion in the anthropogenically stressed Hooghly estuary and DOC influx via mangrove leaf litter leaching in the mangrove dominated estuaries of Indian Sundarbans estuaries (Dutta et al., 2021, Ray et al., 2018)."*

Line 570: why 6800 µatm threshold?

*Response: The 6800µatm threshold was chosen simply based on the nature of the plotted data, which showed a clear breaking point. The plot showed completely different regimes between <6800 µatm and >6800 µatm conditions. While it is possible to perform a change point analysis to statistically determine thresholds with higher precision, the outcome is likely to be very similar. We are not suggesting that 6800 µatm is a quantitatively significant value to consider with respect to diverging estuarine biogeochemical behaviours, but rather, make the case that by splitting the dataset with this arbitrary threshold we can observe different trends. We have softened our description of this relationship to sound less definitive and more exploratory. Additionally, in the revised manuscript we have used PCA to find out the major controlling factors for carbon dynamics of the Indian estuaries.*

Line 579: further explain "a decrease of aerobic bacterial activity with increasing DOC"

*Response: We mean that the increasing DOC load may inhibit bacterial respiration and decreases the $CO_2$ production rate. We have added this clarification to the manuscript.*

Line 616: further explain "freshwater mixing is not the major driver of POC", as it shows lower salinity with higher POC and 13C values.

*Response: Yes, true. Despite low salinity being integrated with higher POC and $\delta^{13}C_{POC}$ values, the relationship between salinity POC and $\delta^{13}C_{POC}$ are not significant (Fig. S9). This indicates mixing between marine and freshwater is not controlled primarily by the estuarine POC dynamics. We have clarified the statement in the revised manuscript.*

Line 668-671: more direct evidence is needed to evaluate anthropogenic impact on $pCO_2$ rather than population density. For example, anthropogenic $pCO_2$ is 100 µatm out of total $pCO_2$ 400 µatm in Estuary A, whereas anthropogenic $pCO_2$ is 200 µatm out of total $pCO_2$ 1000 µatm in Estuary B.

*Response: Yes, we agree that more direct evidence is needed to evaluate the anthropogenic impact on $pCO_2$ rather than population density. But no comprehensive information was available on the degree of anthropogenic discharges in the Indian major estuaries and that forced us to use population density as a proxy for anthropogenic inputs. Although % contribution of anthropogenic $pCO_2$ inputs in total $pCO_2$ (as you suggested) is the best quantitative way to evaluate the importance of anthropogenic inputs, indeed we don't have enough information to estimate the same. This may be considered as a future research scope. For this reason, we have de-emphasized the discussion of anthropogenic inputs by changing this section into a paragraph combined with a new "natural and anthropogenic sources" section. By not having a stand-alone section about anthropogenic inputs we hope that this limitation is alleviated.*

Line 686: where is "Fig. 21"?

*Response: The "Fig. 21" will be corrected as "Fig. S15" (please see supplementary file) in the revised manuscript.*

Line 695-696: wrong statement "nitrification plays crucial role in increasing pH"

*Response: Yes, it was a typographical error. In the revised manuscript the statement is modified as "nitrification plays crucial role in decreasing pH".*

Line 699: "unlikely"

*Response: Sorry, it will be "likely". We have modified it in the revised manuscript.*

Line 712-713: details for FCO2

*Response: We have elaborated on it in the revised manuscript.*

Link 862: where is "Table 6"

*Response: Thanks for pointing out the mistake. The "Table 6" was missed while uploading the tables during the submission process which is attached below. We will take care of it while uploading the revised version of the paper.*

*Table – 6: Contribution of Indian estuaries in global estuarine $CO_2$ and $CH_4$ fluxes to the regional atmosphere. Surface area and flux are given in 'km$^2$' and 'Gg yr$^{-1}$' unit.*

| Parameters | Surface area | Total outgassing flux | References |
|---|---|---|---|
| Global estuarine $CO_2$ flux | $1.40 \times 10^6$ | $2.20 \times 10^6$ | Abril and Borges (2004) |
| | $0.94 \times 10^6$ | $1.58 \times 10^6$ | Borges (2005) |
| | $0.94 \times 10^6$ | $1.17 \times 10^6$ | Borges et al. (2005) |
| | $0.94 \times 10^6$ | $1.32 \times 10^6$ | Chen and Borges (2009) |
| | $1.10 \times 10^6$ | $0.99 \times 10^6$ | Borges and Abril (2012) |
| Mean $CO_2$ flux | $1.06 \times 10^6$ | $1.45 \times 10^6$ | Present study |
| Indian estuarine $CO_2$ flux | $2.70 \times 10^4$ | $9.72 \times 10^3$ | Present study |
| Contribution by Indian estuaries | 2.54% | 0.67% | Present study |
| Global estuarine $CH_4$ flux | $1.40 \times 10^6$ | $1.05 \times 10^3$ | Bange et al. (2004) |
| | $1.40 \times 10^6$ | $1.30 \times 10^3$ | Upstill-Goddard et al. (2000) |
| | $1.40 \times 10^6$ | $2.40 \times 10^3$ | Middelburg et al. (2002) |
| | $1.10 \times 10^6$ | $6.60 \times 10^3$ | Borges and Abril (2012) |
| Mean $CH_4$ flux | $1.33 \times 10^6$ | $2.84 \times 10^3$ | Present study |
| Indian estuarine $CH_4$ flux | $2.70 \times 10^4$ | 3.27 | Present study |
| Contribution by Indian estuaries | 2.54% | 0.12% | Present study |

**Tables and Figures**

Table 1: add coordinates, references

*Response: Exact sampling coordinates for all the estuaries are not mentioned in the paper wherefrom the data is derived. Based on the availability we have updated it in the revised manuscript. But references have been added in the revised manuscript.*

Table 2: are they annual averaged numbers? Standard deviation?

*Response: In table – 1, the degree of precipitation and annual discharge are presented as annual average but standard deviations are not available from where the data was picked up. Regarding tidal amplitude, it is not clear in the relevant paper (Sarma et al., 2012) whether it is annual average or not. So, we are unable to clarify it. The other parameters like number of dams, catchment area and population density are not linked with the annual average.*

Table 5: confusing table, please improve

*Response: Sorry, we have removed table – 5 from the revised manuscript as the DIC and DOC export flux data which are the baseline data for the table are for the riverine export flux not for the estuarine export flux.*

Fig. 1: why only zoom in two estuaries? Instead display C3 and C4 plants area, population density is more important to be visualized.

*Response: The reason for zooming into the Hooghly-Sundarbans estuaries in Fig. 1 is mentioned earlier. We agree population density is a part of C biogeochemistry, but the distribution of C3 and C4 is one of the major sources of terrestrial C that ultimately discharges in the river, eventually to the estuary and continental shelf. Moreover, some of the estuaries (like the estuaries of Sundarbans) have very limited anthropogenic influence. Based on that,*

*we belief it is better to present distributions of C3 and C4 plants rather than population density. I hope the justification will be acceptable to the reviewer.*

Fig. 2 - Fig. 6: cannot distinguish that data between dry and wet, standard deviation needed.

*Response: We have revised the figure in the revised manuscript but presentation of standard deviation is not possible in all the cases due to the scarcity of replicate datasets. We have added standard deviation where possible in the revised manuscript.*

Fig. 6: estuarine export fluxes values should be river-borne C, the figure is unnecessary if only two components are evaluated.

*Response: Yes, we have removed the figure from the revised manuscript.*

For all supplementary figures: there is no spatial information, reason why exclude several estuarine data, the number of observations are too small, standard deviations? Data interpretation seems unconvincing due to potential data manipulation.
**Citation**: https://doi.org/10.5194/bg-2022-200-RC1

*Response: Yes, we agree with the reviewer. For complete understanding, spatial coverage is very important in the case of estuary as elemental biogeochemistry changes along estuarine salinity gradient. In this regard, although some of the Indian estuaries are spatially covered, only their mean value is presented in the paper during data presentation. Some of the estuaries like the Hooghly-Sundarbans system are spatially well discussed. We have added some information on it in the revised manuscripts.*

*We have included a PCA in the revised manuscript now without eliminating a single data to find out major driving forces.*

*Yes, Indian estuaries are not been very thoroughly studied to date. So, until now there is a scarcity of published data, which makes it difficult to present standard deviations in all cases. But in the revised manuscript we have added standard deviation wherever possible.*

*We have tried to improve data interpretation in the revised manuscript wherever possible.*

---

## Author Comment (AC2)

**Reviewer** – 2:**

This study compiled the data of carbon dynamics in the estuaries of India and overviewed the regulating factors of carbon dynamics and the contribution of Indian estuaries on global carbon budgets. This approach is helpful to understand the role of continental estuaries on global carbon cycling. However, this version of manuscript contains major concerns which the authors have to improve before publication. In particular, I think the analytical approach and the interpretation of data should be revised substantially.

**Response:** Thank you for the encouraging comments on the manuscript. We have addressed the concerns raised by both reviewers, which has substantially improved the manuscript.

First, the authors discussed the mechanism of regulating factors of carbon dynamics mainly based on correlation between carbon and other physicochemical parameters but the results of these analysis were not shown in main Figures. If these analyses are substantially used in discussion section, the main text figures and tables should be restructured according to the main agenda.

**Response:** As we had around 20 figures presenting the relationship between variables in the original draft of our manuscript, we decided to present many of the figures showing correlations as supplementary figures. As suggested by the reviewer, we have restructured the figures and tables so as to support the findings and arguments in the discussion section. We also kept numerous figures in the supplement because the visualization of correlations is not essential to the story; rather, we provide the supplemental figures in case a reader is interested to dive deeper into a particular set of parameters.

In addition, the statistical analysis must pay attention to the multicollinearity of multivariate variables. For example, there is a correlation between river flow and population density, which may have a combined effect on carbon concentrations. I think the author should try some analytical methods such as principal component analysis. Although they mention various regulating factors, it is very difficult to understand from the manuscript what is the key controlling factor. I suggest analysis be extracted and discussed based on the statistical results of the above multivariate analyses.

**Response:** Yes, we agree with the reviewer. We have performed PCA test to find out the major controlling factors for estuarine C biogeochemistry in India and the findings are given below and accordingly we revised the manuscript.

Principal component analysis (PCA) was performed in order to identify major controlling factors for dissolved and particulate C as well as the variability of trace gases (CO2 and CH4) in the major estuaries of India. The PCA was performed for 7 parameters (DIC and its isotopic composition, DOC, POC and its isotopic compositions, pCO2, and CH4 concentration) based on the availability of other parameters (no of dams, population density, precipitation, estuarine area, discharge, catchment area, salinity, %DO, and pH). Moreover, due to the scarcity of data during the dry season, the PCA analysis is restricted only during the wet season. The principal component (PC) with eigenvalues >1 was considered for further analysis and only two factors were identified in this case as given below:

|                        | BB estuaries |       | AS es | stuaries |
|------------------------|--------------|-------|-------|----------|
| Eigen value            | 2.90         | 2.07  | 4.40  | 1.65     |
| Explained variance (%) | 41.48        | 29.65 | 62.83 | 23.58    |
| Cumulative             | 41.48        | 71.11 | 62.63 | 86.41    |
| Variable               | PC1          | PC2   | PC1   | PC2      |
| DIC                    | 0.89         | 0.14  | 0.92  | 0.21     |
| $\delta^{13}$ Cdic     | 0.73         | 0.27  | -0.65 | 0.67     |
| DOC                    | 0.40         | 0.55  | 0.58  | 0.72     |
| POC                    | -0.76        | 0.46  | 0.76  | -0.61    |
| $\delta^{13}$ Cpoc     | -0.82        | -0.35 | 0.92  | -0.21    |
| $pCO_2$                | -0.35        | 0.80  | 0.87  | 0.46     |
| CH4                    | -0.23        | 0.84  | 0.78  | 0.12     |

Table: Results for the Principal Component Analysis of estuarine carbon biogeochemistry of India.

The PC1 accounts for ~41.48% and 62.63% variability for the BB and AS estuaries with strong factor loadings of DIC,  $\delta^{13}C_{DIC}$ , POC, and  $\delta^{13}C_{POC}$  for BB estuaries and DIC,  $\delta^{13}C_{DIC}$ , DOC, POC,  $\delta^{13}C_{POC}$ , pCO2 and CH4 for the AS estuaries. For the BB estuaries, the factor loading might be linked with the biological productivity; however, for the AS estuaries, the factor loading might be due to POC and DOC decomposition and its associated productions of CO2 and CH4. The oxygenated estuary supports aerobic degradation of organic matter producing CO2 and considered to be restrained by anaerobic degradation. However, it should also be noted that CH4 production in the isolated anoxic microhabitats of sinking particulate organic matter, in the well-oxygenated water column, has been observed in the open ocean (see Reeburgh, 2007). The PC2 represents 71.11% and 86.41% of the total variance for the BB and AS estuaries with relatively strong factor loading for DOC, pCO2, and CH4 for the BB estuaries and  $\delta^{13}C_{DIC}$ , DOC, and POC for the AS estuaries. The former might be associated with aerobic and anaerobic degradations of DOC; however, the latter might be linked with the aerobic decomposition of DOC and POC and their impact on the variability of  $\delta^{13}C_{DIC}$ .

Thereafter, Pearson correlation analysis was performed for the factor loadings of the estuarine data with the number of dams, estuarine area, population density, mean annual discharge, catchment area, salinity, pH and %DO. The Pearson correlation matrix for the BB estuaries suggests PC1 is strongly controlled by the area of the estuary while PC2 is controlled by the cumulative interactions between salinity, pH as well as catchment area. For the AS estuaries, despite the PC1 being controlled by the number of dams, catchment area, salinity, and %DO but the PCA analysis failed to highlight any controlling factor for the PC2.

| Variables                         | Dams   | Population
(/km2) | Area   | Discharge | Precipitation | Catchment
area (x
103 km2) | Salinity | %DO    | рН     | PC1    | PC2    |
|-----------------------------------|--------|----------------------|--------|-----------|---------------|----------------------------------|----------|--------|--------|--------|--------|
| Dams                              | 1      | -0.373               | 0.805  | 0.934     | -0.052        | 0.318                            | -0.368   | -0.272 | -0.675 | -0.420 | 0.528  |
| Population(/km2)                  | -0.373 | 1                    | -0.373 | -0.362    | -0.083        | 0.155                            | 0.490    | 0.161  | 0.293  | 0.108  | -0.209 |
| Area                              | 0.805  | -0.373               | 1      | 0.766     | 0.128         | 0.282                            | -0.370   | -0.439 | -0.675 | -0.605 | 0.596  |
| Discharge                         | 0.934  | -0.362               | 0.766  | 1         | 0.169         | 0.194                            | -0.496   | -0.464 | -0.743 | -0.570 | 0.541  |
| Precipitation                     | -0.052 | -0.083               | 0.128  | 0.169     | 1             | -0.402                           | -0.376   | -0.429 | -0.135 | -0.547 | 0.047  |
| Catchment area (x 10 3 |        |                      |        |           |               |                                  |          |        |        |        |        |
| km2)                              | 0.318  | 0.155                | 0.282  | 0.194     | -0.402        | 1                                | -0.181   | 0.030  | -0.273 | 0.465  | 0.720  |
| Salinity                          | -0.368 | 0.490                | -0.370 | -0.496    | -0.376        | -0.181                           | 1        | 0.680  | 0.748  | 0.175  | -0.657 |
| %DO                               | -0.272 | 0.161                | -0.439 | -0.464    | -0.429        | 0.030                            | 0.680    | 1      | 0.848  | 0.516  | -0.539 |
| pH                                | -0.675 | 0.293                | -0.675 | -0.743    | -0.135        | -0.273                           | 0.748    | 0.848  | 1      | 0.412  | -0.741 |
| PC1                               | -0.420 | 0.108                | -0.605 | -0.570    | -0.547        | 0.465                            | 0.175    | 0.516  | 0.412  | 1      | 0.000  |
| PC2                               | 0.528  | -0.209               | 0.596  | 0.541     | 0.047         | 0.720                            | -0.657   | -0.539 | -0.741 | 0.000  | 1      |

Correlation Matrix (Pearson) for the BB estuaries:

Values in bolded digits are statistically significant at alpha = 0.05

Correlation Matrix (Pearson) for the AS estuaries:

| Variables             | Dams   | Population
(/km2) | Area   | Discharge | Catchment
area (x 103
km2) | Salinity | %DO    | рН     | PC1    | PC2    |
|-----------------------|--------|----------------------|--------|-----------|----------------------------------|----------|--------|--------|--------|--------|
| Dams                  | 1      | -0.092               | 0.518  | 0.686     | -0.713                           | -0.554   | -0.890 | 0.550  | 0.821  | 0.224  |
| Population (/km2)     | -0.092 | 1                    | 0.253  | -0.202    | -0.550                           | -0.376   | -0.271 | 0.184  | 0.457  | -0.509 |
| Area                  | 0.518  | 0.253                | 1      | 0.828     | -0.502                           | -0.491   | -0.569 | 0.560  | 0.490  | -0.552 |
| Discharge             | 0.686  | -0.202               | 0.828  | 1         | -0.455                           | -0.380   | -0.645 | 0.696  | 0.412  | -0.353 |
| Catchment area (x 103 |        |                      |        |           |                                  |          |        |        |        |        |
| km2)                  | -0.713 | -0.550               | -0.502 | -0.455    | 1                                | 0.729    | 0.938  | -0.699 | -0.940 | 0.265  |
| Salinity              | -0.554 | -0.376               | -0.491 | -0.380    | 0.729                            | 1        | 0.687  | -0.319 | -0.777 | 0.177  |
| %DO                   | -0.890 | -0.271               | -0.569 | -0.645    | 0.938                            | 0.687    | 1      | -0.685 | -0.935 | 0.100  |
| pH                    | 0.550  | 0.184                | 0.560  | 0.696     | -0.699                           | -0.319   | -0.685 | 1      | 0.501  | -0.531 |
| PC1                   | 0.821  | 0.457                | 0.490  | 0.412     | -0.940                           | -0.777   | -0.935 | 0.501  | 1      | 0.000  |
| PC2                   | 0.224  | -0.509               | -0.552 | -0.353    | 0.265                            | 0.177    | 0.100  | -0.531 | 0.000  | 1      |
|                       |        |                      |        |           |                                  |          |        |        |        |        |

Values in bolded digits are statistically significant alpha = 0.05

The analysis with outliers removed is also very arbitrary. I think the variability of freshwater endmember would cause such outliers. My recommendation is to analyze the effects of mixing and biogeochemical processes in estuaries separately from the determinants of the river endmember values.

**Response:** Yes, we understand the application of the mixing model is a good approach to separate the effects of mixing and biogeochemical processes. However, in our case, we are constrained by the lack of seasonal data for end members of each river which prevents us from applying the mixing model.

**Line comment**

203) I think the compiled dataset is very useful for further studies. Don't you open this via any repository?

**Response:** All data used in the manuscript has been compiled and the data shall be made available either via a repository or will be presented as supplementary material for further use.

208) What kind of statistical analyses did you use? You have to explain the approach.

**Response:** In the original manuscript we used simple regression analysis to test dependency between variables. Based on the suggestions by the reviewers, we have performed 't' test and PCA test (for the wet season only when suitable data is available) and the same is included in the revised manuscript (results of 't' test are given below and results of PCA is detailed in the earlier comments). The Material and Method section was updated accordingly in the revised manuscript.

|                      | Inter-seasona       | l comparison       | Inter-BB & AS comparison |                    |  |  |
|----------------------|---------------------|--------------------|--------------------------|--------------------|--|--|
| Parameters           | BB estuaries | AS estuaries       | Wet Season               | Dry Season         |  |  |
| Salinity             | p = 0.55     | p <0.0001** | p = 0.003**       | p = 0.84    |  |  |
| %D0                  | p = 0.60     | p = 0.31    | p = 0.07          | p = 0.001** |  |  |
| pН                   | p = 0.10     | p = 0.006** | p <0.0001**       | p = 0.012** |  |  |
| DIC                  | p = 0.59     | NA                 | p = 0.008**       | NA                 |  |  |
| $\delta^{13}C_{DIC}$ | p = 0.69     | NA                 | p = 0.000**       | NA                 |  |  |
| DOC                  | p = 0.32     | NA                 | p = 0.45          | NA                 |  |  |
| РОС                  | p = 0.02**   | p = 0.02**  | p = 0.17          | p = 0.06    |  |  |
| $\delta^{13}C_{POC}$ | p = 0.04 **         | NA                 | p = 0.21          | NA                 |  |  |
| pCO 2     | p = 0.07     | NA                 | p = 0.10          | NA                 |  |  |
| FCO 2     | p = 0.41     | NA                 | p = 0.29          | NA                 |  |  |
| CH4                  | p = 0.39     | p = 0.38    | p = 0.03**        | p = 0.11    |  |  |
| FCH4                 | p = 0.05**   | p = 0.27    | p = 0.13          | p = 0.16    |  |  |

Table – Results for statistical 't' test analysis between the mean of available data. The analysis is performed at 95% confidence level.

\*\*Statistically significant at  $\alpha = 0.05$ .

224) You often indicate in this manuscript how large or small by %, is this comparison only rivers for which you have data for both wet and dry seasons? If you are compiling all data, you will have a bias due to the different rivers you are averaging.

**Response:** Yes, the comparison is based on the seasonal variation among BB and AS estuaries. While calculating the average, we have considered all estuaries mixing with Bay of Bengal under the BB estuaries category and all estuaries mixing with the Arabian Sea under AS estuaries category.

Fig. 2-6) This value is average in each estuary? At least, you should show error bars. Is possible, you should show whisker plots.

**Response:** Yes, for the graphical presentation we have used an average value where multiple values are available. As in most cases, multiple data for a particular C parameter are not

available to show whisker plot. The challenge with presenting data in such a manner is that most studies we cite do not include enough data points to visualize in box plots, for example.

In result section) You used "higher" or "lower" terms. These are based on statistical analysis? All comparisons should be based on statistical analyses.

**Response:** No, it is not based on statistical analysis. The term "higher" and "lower" are simply decided based on the magnitude of the data. However, as suggested by the reviewer, we have performed the 't' test to check if the data of two seasons/estuaries is significantly different or not and the details of the 't' test results are given earlier, and the same will be included in the revised manuscript.

245) Basically, outliers should not be arbitrarily removed. It would be interesting to discuss the factors that cause freshwater endmembers to vary.

**Response:** We have performed PCA (results shown earlier) now in place of simple regression analysis. We have added it to the revised manuscript. Sufficient information and values are not available on freshwater end members of respective rivers.

253, 266) Is the average also higher than in estuaries around the world?

**Response:** The mean DIC and DOC concentrations of the Indian estuaries were calculated as 1780 and 379  $\mu$ mol L-1, respectively. If we compare the mean values with global estuarine DIC and DOC (table 2 & 3), the values for the Indian estuaries are intermediate between low and high values reported for the global estuaries. We have modified the content in the revised manuscript to avoid any ambiguity in the paper.

256) Here, "peak" may not be suitable. Higher-lower or heavier-lighter are often used.

**Response:** Thanks for the suggestion, we have modified it in the revised manuscript.

265) for dry season?

**Response:** Yes, it is during the dry season. We have clarified it in the revised manuscript.

290) unit

**Response:** The unit "nM" is included in the revised manuscript.

327) Rainfall dilute riverine DIC?

**Response:** Yes, Krishna et al. (2019) showed the same for the Indian estuaries.

331) These values are averages with the broad salinity range? It is difficult to differentiate the mixing effect from the freshwater endmember variability.

**Response:** Regarding Indian estuaries, although two end-member mixing model was previously applied for some of the estuaries like the Godavari, Hooghly by Bouillon et al. (2003), Samanta et al. (2015) and Dutta et al. (2019, 2021) to differentiate mixing effect from the biogeochemical and anthropogenic impacts. We have included some of the major findings

from the mixing model study by them in the revised manuscript. But, despite the availability of marine end member data (the Arabian Sea and the Bay of Bengal), seasonal freshwater end member data for most of the estuaries are not available to date which restricts us to use the mixing model to differentiate mixing effect from other effects.

332) BB and AS use different fitting curves, but aren't they just different ranges of precipitation? I think it would be more general if the same relationship equation could be used to explain the difference.

**Response:** We have performed PCA (results shown earlier) now in place of simple regression analysis. The PCA showed no significant correlation with precipitation. We have added it in the revised manuscript.

392) Also degassing of CO2?

**Response:** Yes, we agree. CO2 outgassing is also responsible for DIC removal depending upon gas transfer velocity and air-water partial pressure gradient. This has been clarified it in the revised manuscript.

402) Rivers with large population densities may have large dilution of river flow. Multivariate analysis may be effective.

**Response:** We have performed a PCA to test the same and it has added to the revised manuscript.

420) The relationship between precipitation and DIC should also be discussed comprehensively. DIC supply due to carbonate weathering may dominate in rivers with low precipitation.

**Response:** We have performed PCA (results shown earlier) that showed no significant correlation with precipitation. We have updated it in the revised manuscript.

468) This paragraph is redundant because it is a general statement.

**Response:** We have removed it from the revised manuscript.

492) Fig. 12?

**Response:** It is a typographical error. We have corrected it in the revised manuscript.

496) p=0.06 is not significant

**Response:** We have corrected it in the revised manuscript.

521) There may be a combined effect of river discharge and population density.

Response: Yes true. To test the same PCA analysis was added in the revised the manuscript.

536) This may also be an effect of multicollinearity.

**Response: Agree. To test the same PCA analysis was added in the revised the manuscript.**

578) Splitting a fitting line is arbitrary if there is no meaning in  $6800 \ \mu atm$ . The influence of other variables should be considered.

**Response:** We have performed PCA (results shown earlier) now in place of the simple regression analysis that used in the pre-revised manuscript. We have added it to the revised manuscript.

595) Without an OM source mixing model (using more than 2 variables), it is difficult to discuss the contribution of each carbon source. For example, d13C value of -24~-19‰ can be explained by the mixing between C3 and C4 without marine origin.

**Response:** Yes, it's true that the OM source mixing model needs to solve to discuss the contribution of individual POC sources. We mentioned only about possible POC sources here.

630) I think the quantity and quality of POC cause the decomposition and O2 consumption rather than isotopic fractionation. Isotope fractionation doesn't happen that often with degradation (if it did, POC would be noticeably reduced).

**Response:** Thank you for the information. We have removed it from the revised manuscript. In the revised manuscript we have used PCA to find out the major controlling factors.

**653) Why? It is interesting.**

**Response:** We have performed PCA (results shown earlier) now in place of simple regression analysis. The PCA clearly indicates no potential impact of discharge in both BB and AS estuaries during the wet season. We have added the following interpretation of why this relationship is not significant: "...perhaps due to a complex interplay between nutrient inputs, primary productivity, and microbial respiration that complicate conservative mixing of  $CO_2$ ."

Citation: https://doi.org/10.5194/bg-2022-200-RC2